# Identification of core genes in the extracellular matrix and the regulatory mechanisms of the immune microenvironment in idiopathic pulmonary fibrosis using WGCNA and machine learning methods

Man Wang[iD], Lu Liu, Yang Liu, Shihuan Yu[iD]*

Department of Respiratory Medicine, The First Affiliated Hospital of Harbin Medical University, Harbin, Heilongjiang Province, China

* yushihuan2000@126.com

## Abstract

### Objective

This research aims to detect genes associated with the extracellular matrix (ECM) in idiopathic pulmonary fibrosis (IPF) using bioinformatics techniques and investigate their relationships with immune infiltration, with the goal of identifying new diagnostic and therapeutic targets for IPF.

### Methods

The study employed a combination of differential expression analysis, weighted gene co-expression network analysis (WGCNA), and various machine learning algorithms to screen for characteristic genes. Gene set enrichment analysis (GSEA), gene ontology (GO), and Kyoto Encyclopedia of Genes and Genomes (KEGG) were utilized to evaluate relevant biological functions and pathways. Additionally, the analysis of immune cell infiltration was conducted to assess the disease's immune status and the correlations between genes and immunity.

### Results

IPF is strongly linked to pathways such as ECM organization and immune response, with differentially expressed genes primarily involving signal pathways related to collagen deposition in the extracellular matrix. A total of 1,193 ECM-related genes associated with IPF were identified, and 94 differentially expressed ECM-related genes were further screened compared to the normal control group. Through machine learning approaches, three key genes—*BAAT*, *COMP*, and *CXCL13*—were pinpointed. These genes are closely tied to the onset, progression, and immune

**Data availability statement:** All relevant data are within the manuscript and its Supporting Information files.

**Funding:** This study was supported by Wu Jieping Medical Foundation, Project Number 320.6750.2021-04-22, awarded to Principal Investigator: Dr. Shihuan Yu.

**Competing interests:** The authors have declared that no competing interests exist.

processes of IPF, and clustering analysis based on them can reveal distinct disease states and changes in immune cell infiltration patterns.

## Conclusion

*BAAT, COMP,* and *CXCL13* may serve as potential therapeutic targets for slowing the progression and preventing the exacerbation of IPF. Moreover, monocytes demonstrate consistent infiltration patterns across the disease group, control group, and various subgroups, indicating their potential significance in the development of IPF.

## Introduction

Idiopathic pulmonary fibrosis (IPF) is a fatal interstitial lung disease with a high incidence among the elderly, characterized by progressive respiratory function deterioration and high mortality, presenting significant challenges in clinical management [1]. Research indicates that genetic susceptibility (such as MUC5B promoter polymorphism) and environmental exposure (dust, pollutants) interactively drive the onset of the disease, with extracellular matrix (ECM) remodeling [2] and immune microenvironment dysregulation playing a core role. Although existing drugs (nintedanib/pirfenidone) can slow down the progression [3], their efficacy is limited and side effects are significant, making it urgent to achieve breakthroughs in targeted therapy. Therefore, in-depth exploration of the molecular mechanisms and signaling pathways of IPF is crucial for advancing therapeutic strategies.

The pathogenesis of IPF is believed to originate from persistent or recurrent damage to lung epithelial tissue, particularly type II alveolar cells (AT2). This damage initiates abnormal repair responses, leading to fibroblast activation and myofibroblast differentiation. Myofibroblasts are central to excessive ECM accumulation and impaired lung repair, ultimately resulting in tissue scarring, alveolar destruction, and irreversible lung dysfunction. [4]

IPF development is typically viewed as an interaction between genetic and environmental factors. Genome-wide association studies (GWAS) and whole-exome sequencing have identified several genetic risk factors associated with IPF, including polymorphisms in the MUC5B gene promoter (rs35705950) [5], telomere maintenance genes (e.g., *TERT*, *TERC*, *RTEL1*) [6], surfactant protein genes (e.g., *SFTPC*, *SFTPA2*) [7], and immune regulation-related genes. These genetic variations may influence disease progression through multiple pathways, such as altering mucin production, disrupting epithelial repair, interfering with surfactant metabolism, and contributing to IPF via mechanisms like telomere shortening and chromosomal instability. Environmental and occupational exposures [8], including inhalation of wood dust, metal dust, asbestos, silica particles, air pollutants, and organic antigens, combined with pre-existing conditions such as gastroesophageal reflux disease (GERD) and chronic viral infections [9], can directly damage lung tissue, triggering chronic inflammation and fibrotic responses. Additionally, unhealthy habits such as smoking and alcohol consumption are notable risk factors [10]. Smoking introduces harmful

substances that exacerbate oxidative stress and inflammation, accelerating fibrosis, while long-term alcohol use weakens lung repair capabilities, increasing the likelihood of fibrosis.

Although the precise cause of IPF remains elusive, significant progress has been made in elucidating its underlying mechanisms, suggesting the existence of a common fibrotic mechanism. Research demonstrates that various harmful factors can directly damage lung tissues, affecting bronchial epithelial cells, alveolar epithelial cells [11], and vascular endo-thelial cells [12]. This damage disrupts basement membrane structure, leading to cytoskeletal remodeling, increased ECM synthesis, collagen deposition, fibroblast proliferation [13], and transformation of fibroblasts into myofibroblasts, culminat-ing in pulmonary fibrosis [14]. Multiple signaling pathways play critical roles during this process [4]. For instance, damage activates the P53-induced senescence pathway, impairing AT2 cell function and reducing lung repair capacity. Activation of the TGF-β/Smad pathway regulates fibrosis-related gene expression and promotes excessive ECM generation. Enhanced platelet-derived growth factor (PDGF) signaling interacts with PDGF receptors (PDGFR-α and PDGFR-β), activating downstream pathways such as Ras-MAPK, PI3K/Akt, and PLC-γ. Upregulation of the PI3K/Akt/mTOR pathway and abnor-mal Notch signaling facilitate the transformation of fibroblasts and pericytes into myofibroblasts, which are crucial for ECM deposition and tissue remodeling. Excessive reactive oxygen species production and dysfunctions in autophagy and apop-tosis pathways also contribute to epithelial cell damage, fibroblast activation, and ECM remodeling, exacerbating fibrosis.

Currently, IPF diagnosis relies on pulmonary imaging, pulmonary function tests, and pathological evaluations. However, due to difficulties in obtaining pathological samples and the complexity of diagnosis, there is an urgent need for novel bio-markers to enhance diagnostic efficiency and accuracy. Ideal biomarkers should be easily detectable and capable of distin-guishing pulmonary fibrosis from other interstitial lung diseases. Current research focuses on three areas: markers of alveolar epithelial cell injury, ECM remodeling and fibrosis-related markers, and immune cell and inflammatory regulation markers. In ECM remodeling, key research directions include ECM deposition, stiffness, endoplasmic reticulum stress, inflammatory responses, and immune regulation. Excessive ECM deposition is a hallmark of IPF, with changes in matrix metalloproteinases (MMPs) and tissue inhibitors of metalloproteinases (TIMPs) playing a pivotal role. Altered ECM com-ponents affect fibroblast transcriptional profiles and establish positive feedback loops, promoting fibrosis. Matrix stiffening [15], characterized by abnormal collagen cross-linking, sustains pulmonary fibrosis. Studies suggest that targeting mecha-nosensitive signaling pathways in myofibroblasts could offer potential therapeutic strategies. Endoplasmic reticulum stress (ERS) [16] refers to cellular imbalance between protein synthesis demands and processing capacity. ERS activates the unfolded protein response (UPR) to restore homeostasis but can lead to apoptosis under severe conditions. In IPF, UPR stimulates pro-fibrotic mediators such as transforming growth factor-β (TGF-β), platelet-derived growth factor (PDGF), and CXC chemokine 12 (CXCL12) [17]. While the exact role of inflammation in IPF pathogenesis remains unclear, neutrophils recruited to injured sites release pro-inflammatory cytokines and neutrophil elastase (NE), intensifying fibrosis. During acute exacerbations, elevated interleukin-17 (IL-17) and interleukin-23 (IL-23) levels indicate IL-23's significance. Monocytes and macrophages drive fibrosis through excessive alveolar cell repair, with secreted CCL2 and colony-stimulating factor (M-CSF/CSF1) potentially contributing to fibrosis formation. Recent studies show that immune cells in lung tissue predict IPF severity and progression, offering prognostic value. [18]

This study aims to screen IPF-specific biomarkers through bioinformatics, and construct a multi-parameter monitoring system to guide early diagnosis and personalized treatment. Integrating biomarkers with clinical data can precisely assess disease stages, providing a theoretical basis for the development of novel therapies targeting ECM remodeling and signal pathway regulation.

## 2 Methods

### 2.1 Data Source and Processing

The Gene Expression Omnibus database from the National Center for Biotechnology Information in the United States was retrieved to search for datasets of IPF. Ultimately, two datasets derived from lung tissue samples of patients with idiopathic pulmonary fibrosis

(IPF) and healthy control individuals were screened and acquired: GSE150910 (an RNA-seq dataset) and GSE70866 (a microarray dataset). Subsequently, further analyses were performed on these datasets [19]. The two datasets contained the gene expression information of IPF patients (IPF) and healthy individuals (Control). GSE150910 was used as the training set for differential analysis, WGCNA, etc. to identify key genes, and GSE70866 was used for the expression verification of core genes.

For data processing, the expression matrix of the data was used to remove rows (genes) and columns (samples) with a missing value ratio greater than 50%, and then the impute.knn function of the R software package impute was used to complete the missing values and perform log2 conversion on the data.

## 2.2 Analysis of Differences and Functional Enrichment Analysis

We employed the R package limma (version 3.40.6) [20] to conduct differential analysis for identifying differentially expressed genes in the lung tissues of the IPF group versus the Control group, with the cutoff value set as |log2 fold change| > 1. Regarding the functional enrichment analysis of gene sets, we utilized the KEGG rest API to acquire the latest gene annotations of KEGG Pathway and the GO annotations of genes in the R package org.Hs.e.g.,db (version 3.1.0). Taking these as the background, we mapped the genes onto the background set and carried out enrichment analysis using the R package clusterProfiler (version 3.14.3) to obtain the results of gene set enrichment.

## 2.3 WGCNA

Taking the gene expression profile as an example: Firstly, we utilized the gene expression profile to calculate the MAD (Median Absolute Deviation) of each gene respectively, and eliminated the top 50% of genes with the smallest MAD. The outlier genes and samples were removed using the goodSamplesGenes method of the R software package WGCNA. Further, WGCNA was used to construct a scale-free co-expression network. Specifically, both the Pearson's correlation matrices and the average linkage method were performed for all pairwise genes. Then, a weighted adjacency matrix was constructed using a power function $A\_mn = |C\_mn|^\beta$ (C_mn = Pearson's correlation between Gene_m and Gene_n; A_mn = adjacency between Gene m and Gene n). β was a soft-thresholding parameter that could emphasize strong correlations between genes and penalize weak correlations. After choosing the power of 3, the adjacency was transformed into a topological overlap matrix (TOM), which could measure the network connectivity of a gene defined as the sum of its adjacency with all other genes for network generation, and the corresponding dissimilarity (1 – TOM) was calculated. To classify genes with similar expression profiles into gene modules, average linkage hierarchical clustering was conducted according to the TOM-based dissimilarity measure with a minimum size (gene group) of 5 for the genes dendrogram. The sensitivity was set to 3. To further analyze the module, we calculated the dissimilarity of module eigen genes, chose a cut line for the module dendrogram and merged some modules. Additionally, we also merged the modules with a distance less than 0.25, and ultimately obtained 18 co-expression modules. The four modules, namely darkquoise, darkgreen, darkgary, and greenyellow, which were the most relevant to IPF, were used for further analysis. Using the MM threshold of 0.5 and the GS threshold of 0.1, a total of 1,423 genes were obtained [21].

## 2.4 Target genes are screened by machine learning

To obtain the target genes further, we employed four algorithms, namely LASSO, RF, SVM-RFE, and XGBOOST [22–25], to conduct further dimensionality reduction processing on the gene set. The overlapping genes obtained from the four algorithms were regarded as the target genes.

## 2.5 Gene set enrichment analysis(GSEA)

We obtained the GSEA software (version 3.0) from the GSEA website [26] and classified the samples in accordance with the expression level of the target genes into high-expression groups (≥50%) and low-expression groups (<50%).

Moreover, we downloaded relevant pathway sets from the Molecular Signatures Database for the evaluation of relevant pathways and molecular mechanisms.

### 2.6 Immune infiltration analysis

For the immune infiltration analysis, it was achieved through ImmuCellAI [27]. ImmuCellAI can evaluate the abundances of 24 immune cells in human samples. We visualized the obtained abundances of immune cells and evaluated the relationship between the target gene and immune cells.

### 2.7 Consistent clustering

The clustering analysis was conducted using ConsensusClusterPlus [28]. Agglomerative pam clustering with a search-related distance of 1 was employed, and 10 repeated samplings of 80% of the samples were performed. The optimal number of clusters was determined using the empirical cumulative distribution function graph.

### 2.8 Statistical analysis

All computations and statistical analyses were conducted using R software and Sangerbox 3.0 [29]. A P value of less than 0.05 was regarded as statistically significant.

### 2.9 Ethics approval and consent to participate:Not applicable

## Results

### 3.1 Analysis of differentially expressed genes between IPF and control samples

To assess the differences between normal and IPF samples, a total of 102 normal samples and 102 IPF samples from the GSE150910 dataset were analyzed. This analysis identified 867 upregulated genes and 672 downregulated genes. Volcano plots were generated to visualize the distribution of these genes based on log2 fold change (log2FC) and false discovery rate (FDR) (Fig 1A). The heatmap depicting the expression patterns of these differentially expressed genes is presented in (Fig 1B).

### 3.2 Functional correlation analysis

To gain deeper insights into the potential biological functions of DEGs associated with IPF, we conducted Gene GO and KEGG enrichment analyses. The GO biological process enrichment analysis revealed that these DEGs were predominantly involved in processes such as tissue development, multicellular regulation, cell adhesion, and extracellular matrix formation (Fig 2A). In terms of cellular components, the DEGs were largely associated with collagen-rich extracellular matrices and cell membrane structures (Fig 2B). Regarding molecular functions, they were primarily engaged in activities related to signaling receptor activation, receptor-ligand interactions, and structural components of the extracellular matrix (Fig 2C).KEGG indicated that these DEGs were significantly enriched in pathways including neuroactive ligand-receptor interaction, cytokine-cytokine receptor interaction, ECM-receptor interaction, and renin secretion (Fig 2D). These findings suggest that the extracellular matrix plays a pivotal role in the pathogenesis of IPF.

### 3.3 The establishment of the WGCNA framework screened out the core genes of IPF

To explore co-expression patterns and identify co-expression modules, we employed WGCNA. Hierarchical clustering of the samples was performed using Euclidean distances derived from log-transformed RNA-seq fractional counts (Fig 3A), and the resulting dendrogram indicated minimal outliers in the sample aggregation process. To construct a scale-free network topology, an appropriate soft threshold was determined, converging at approximately 5 for IPF samples (scale-free

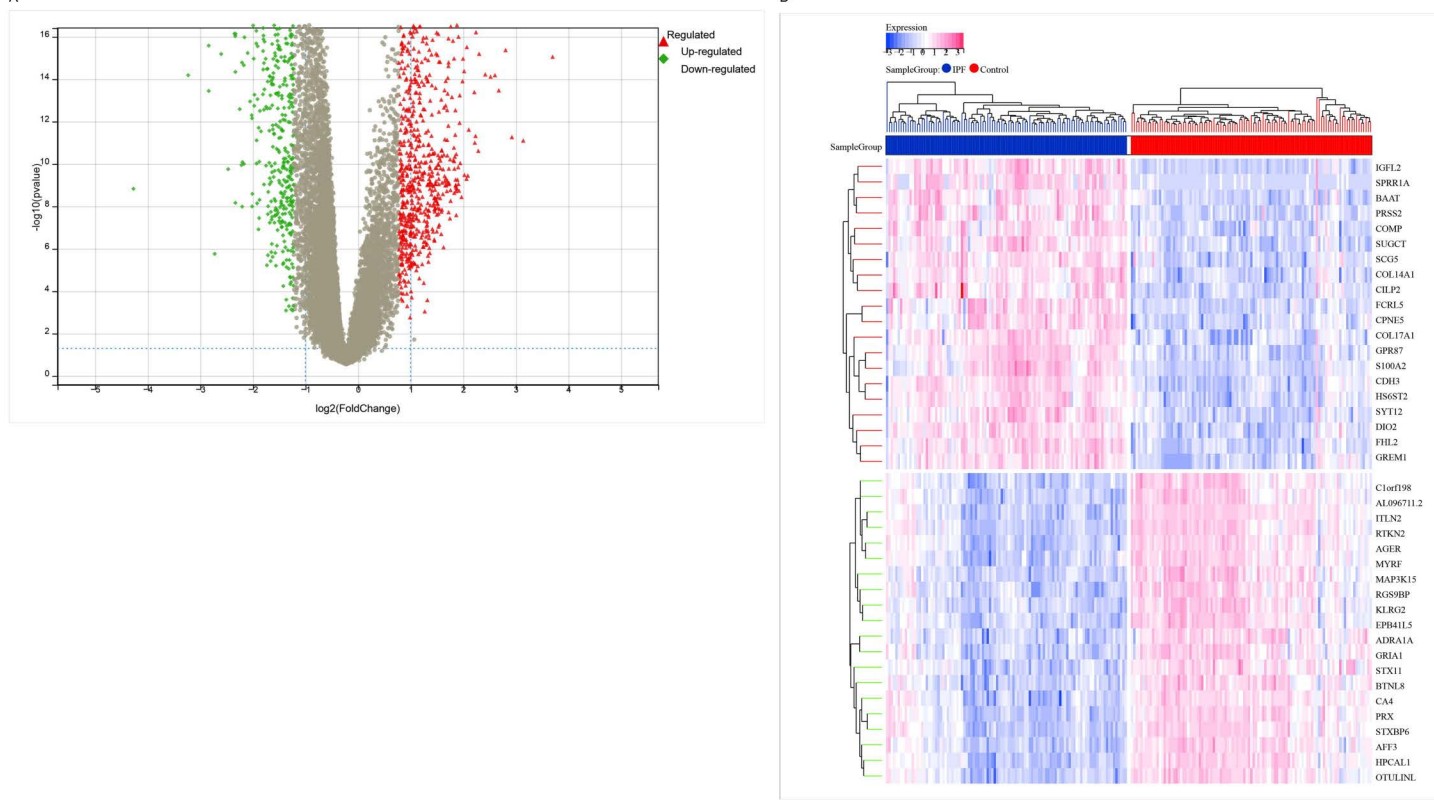

**Fig 1. Differentially expressed genes between idiopathic pulmonary fibrosis and normal samples. (A)** Volcano plot of the GSE150910 dataset with screening criteria (|log2FC|>0.585; FDR<0.05). **(B)** Heatmap of the differentially expressed genes.

R²=0.87) (Fig 3B,C). In the GSE150910 dataset, WGCNA identified 18 distinct modules (Fig 3D,E), each represented by a unique color. The relationship between these modules and disease status was evaluated using a Spearman correlation coefficient heatmap, revealing key modules and their interactions (Fig 3F). Further analysis showed that four modules— dark cyan, dark green, dark gray, and yellow-green—were most significantly associated with IPF and were selected for in-depth investigation. By setting thresholds of module membership (MM) ≥ 0.5 and gene significance (GS) ≥ 0.1, a total of 1,423 genes strongly correlated with IPF were identified. Additionally, scatter plots illustrating the correlation between the four module genes and IPF confirmed their high association with IPF (Fig 3G,H,I,J).

### 3.4 Functional analysis of DEGs

Functional analysis of DEGs indicated that they are likely closely associated with extracellular matrix (ECM) functions. Consequently, 1193 ECM-related genes (ERGs) were retrieved from the GSEA database. By intersecting DEGs with genes obtained from WGCNA and ERGs, we identified 94 IPF-related ECM-related differential genes (ERDEGs)(Fig 4A).

GO enrichment analysis of the IPF-related ECM-related differentially expressed genes showed that their functional annotations were predominantly related to cellular responses to chemical stimuli and organic substances(Fig 4B).

KEGG pathway enrichment analysis revealed that these IPF-related ERDEGs were primarily enriched in pathways such as vascular smooth muscle contraction, focal adhesion, and ECM-receptor interaction.(Fig 4C).

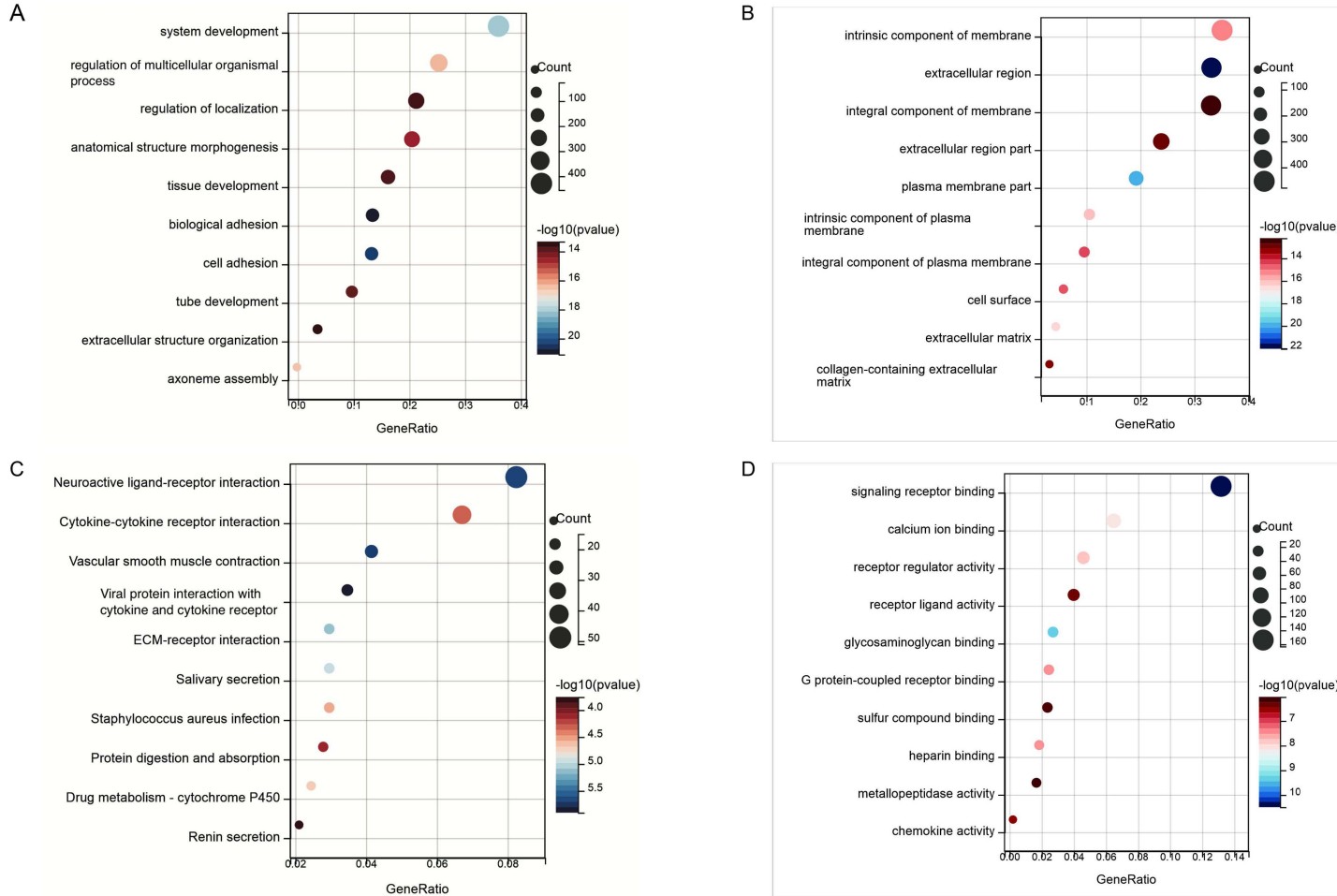

**Fig 2. GO and KEGG pathway analyses of differentially expressed genes in patients with idiopathic pulmonary fibrosis compared to healthy controls.** (A)Biological Process (BP) Analysis; **(B)** Cellular Component (CC) Analysis; **(C)** Molecular Function (MF) Analysis;(D)Kyoto Encyclopedia of Genes and Genomes (KEGG) pathway analyses.

### 3.5 Four machine learning methods were used to further screen 94 ERDEGs

Validated machine learning algorithms, including LASSO, SVM-RFE, RF, and XGBOOST, were employed to identify key characteristic genes associated with IPF-related extracellular matrix (ECM)-related differentially expressed genes (ERDEGs). Specifically, 18 characteristic genes were identified using the LASSO algorithm(Fig 5A, B), 10 using the SVM-RFE algorithm(Fig 5C), 20 biomarker genes were identified using the XGBOOST algorithm(Fig 5D), and 20 using the RF algorithm(Fig 5E, F).

The ROC curve analysis demonstrated that all four machine learning models exhibited robust predictive performance (Fig 6A, B, C, D).The characteristic genes obtained from the four machine learning methods were cross-analyzed, resulting in the selection of only three overlapping genes: *BAAT*, *COMP*, and *CXCL13*(Fig 6E). A Venn diagram illustrated the intersection of characteristic genes identified by these four algorithms. These overlapping genes represent the target genes associated with IPF-related ECM-related ERDEGs. Compared to the control group, the IPF group showed significantly higher expression levels of these three hub genes, as confirmed by the ROC curve analysis, which indicated good predictive performance.

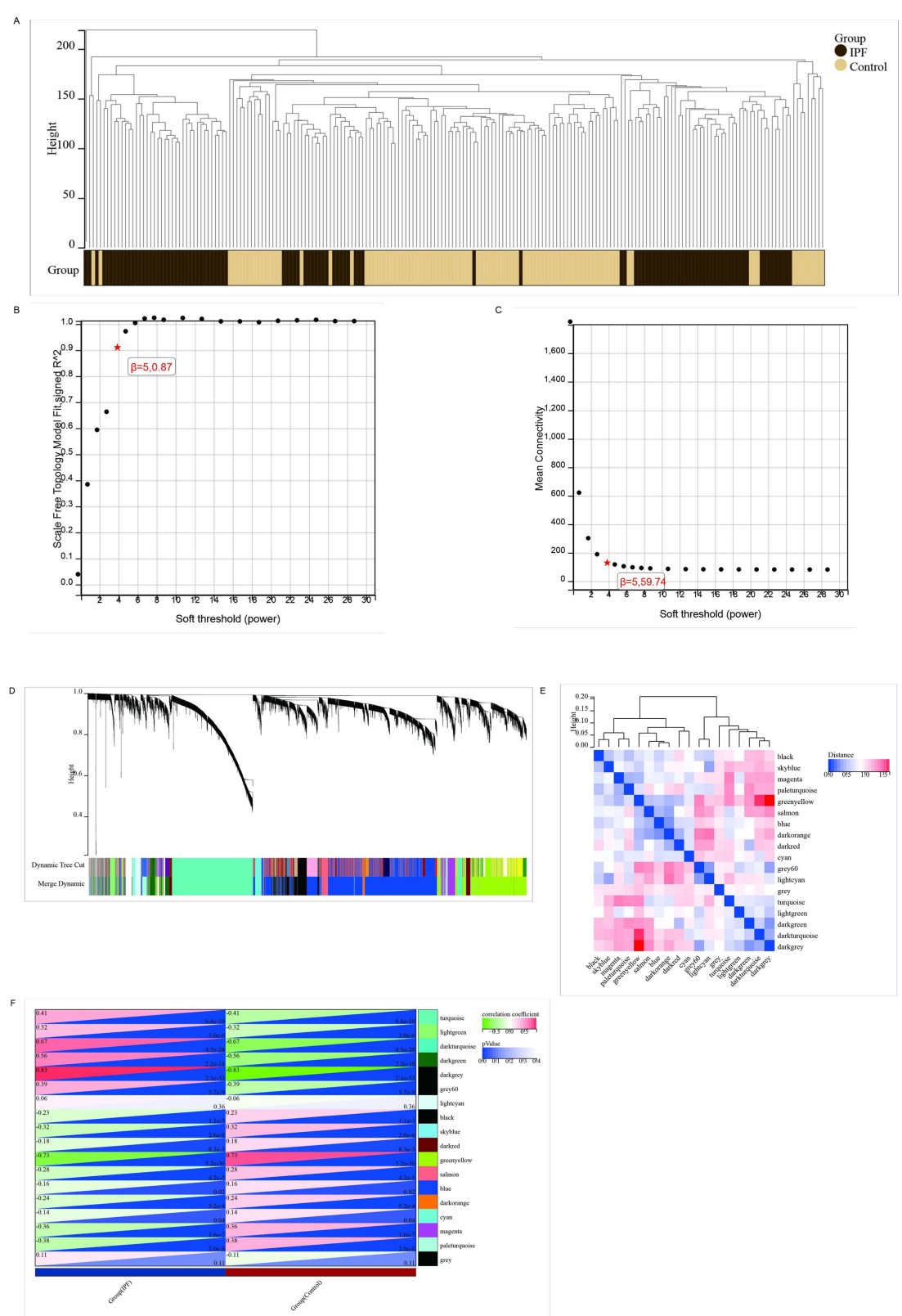

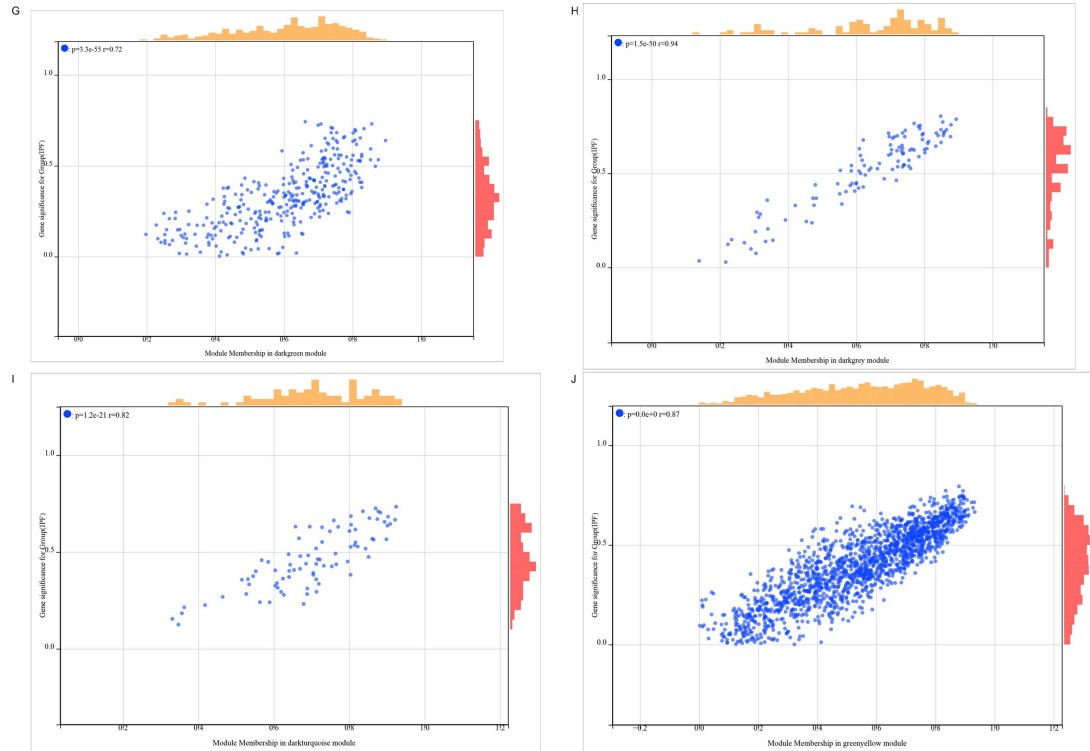

**Fig 3. Identification of IPF-associated gene modules via WGCNA. (A)** Outlier sample detection; **(B, C)** Soft-threshold selection; **(D, E)** Generation of 18 co-expression modules; **(F)** Module-IPF correlation analysis; **(G-J)** Scatter plots depicting correlations between IPF and the darkgreen **(G)**, darkgrey **(H)**, darkturquoise **(I)**, and greenyellow (J) modules.

### 3.6 Single-gene analysis

After intersecting the 94 IPF-related ECM-related ERDEGs using the four machine learning methods, three hub genes (*BAAT*, *COMP*, *CXCL13*) were identified. Their expression levels were validated across three independent datasets. Notably, all three genes exhibited elevated expression in IPF samples compared to the control group(Fig 7A). Additionally, the diagnostic efficacy of these three genes was evaluated in the three datasets. All three genes demonstrated substantial diagnostic value in IPF, with area under the curve (AUC) values exceeding 0.9(Fig 7B,C,D).

A systematic assessment of the functional roles of the *BAAT*, *COMP*, and *CXCL13* genes in IPF was conducted via single-gene gene set enrichment analysis (GSEA) (Fig 8). As depicted in, the top five positively and negatively correlated pathways of the three genes exhibited significant functional associations. The specific analysis results are as follows:

Functional characteristics of the *BAAT* gene: This gene displayed a strong correlation with metabolic pathways. Among the positively correlated pathways, primary bile acid biosynthesis (Primary Bile Acid Biosynthesis, ES = 0.7647, NP < 0.001) and taurine metabolism (Taurine and Hypotaurine Metabolism, ES = 0.6825, NP = 0.026) both manifested extremely significant enrichment (Fig 8A). It is worth noting that the top five positively correlated pathways are as follows: primary bile acid biosynthesis (ES = 0.7647, NP < 0.001); biosynthesis of unsaturated fatty acids (ES = 0.5670, NP = 0.019); biosynthesis of glycosaminoglycan – heparan sulfate (ES = 0.6041, NP = 0.016); taurine and hypotaurine metabolism (ES = 0.6825, NP = 0.026); histidine metabolism (ES = 0.5194, NP = 0.058). The analysis of negatively correlated pathways indicated that endocytosis (Endocytosis, ES = −0.3992, NP = 0.002) and natural killer cell-mediated cytotoxicity (ES = −0.5098, NP = 0.012) were statistically significant (Fig 8B), suggesting that *BAAT* might participate in the IPF process by regulating immune responses.

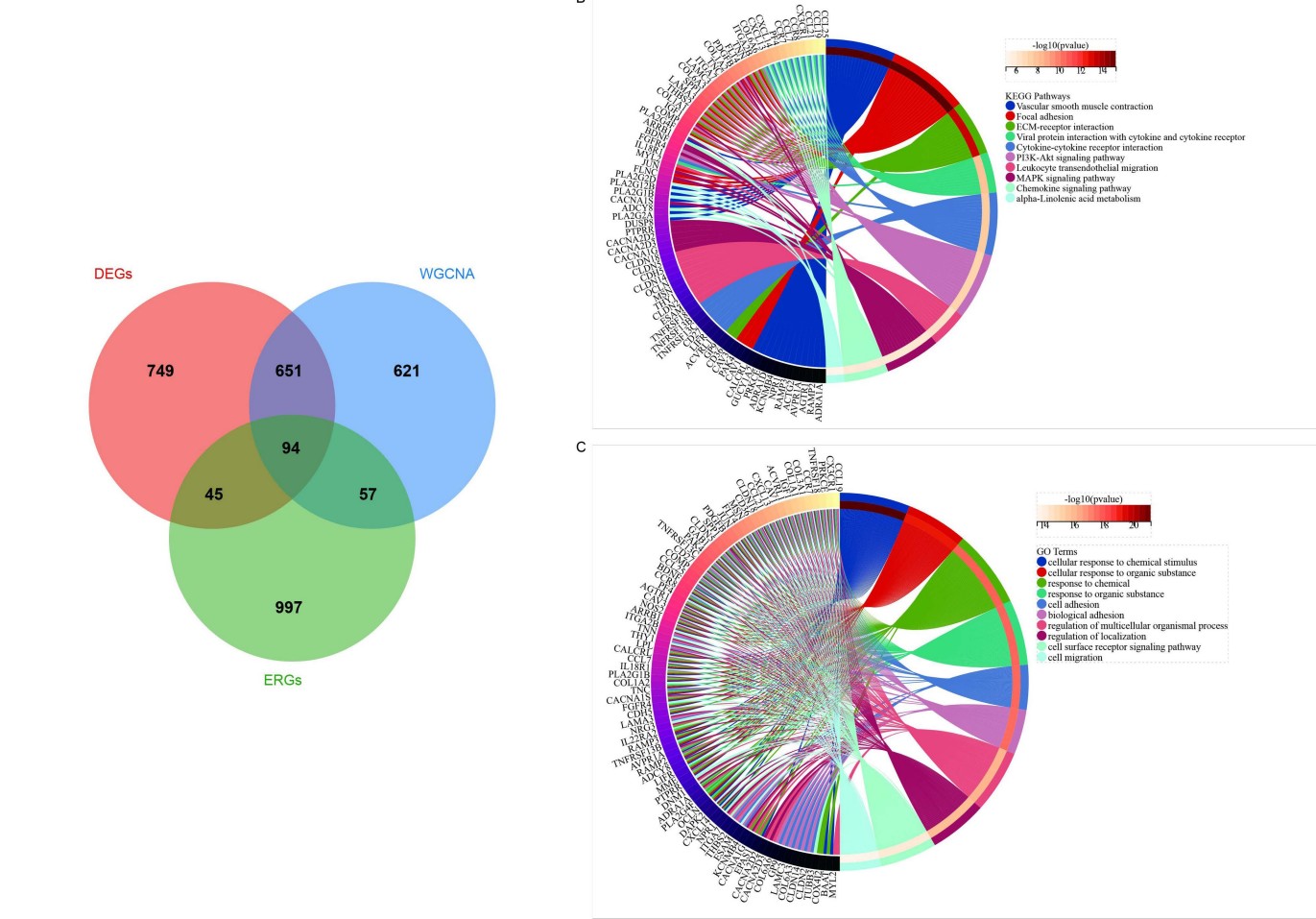

**Fig 4. Screening and functional profiling of extracellular matrix (ECM)-related differentially expressed genes (ERDEGs) in IPF. (A)** ERDEGs screening; **(B)** GO enrichment; **(C)** KEGG pathway analysis.

Pathway interactions of the *COMP* gene: This gene was actively involved in energy metabolism pathways such as the citrate cycle (Citrate Cycle (TCA Cycle), ES=0.4632, NP=0.036) (Fig 8C), and presented a significant negative correlation with the PPAR signaling pathway (ES=−0.5307, NP=0.008). Specifically, the positively correlated pathways are as follows: primary bile acid biosynthesis (ES=0.7647, NP<0.001); biosynthesis of unsaturated fatty acids (ES=0.5670, NP=0.026); taurine and hypotaurine metabolism (ES=0.6825, NP=0.022); biosynthesis of glycosaminoglycan – heparan sulfate (ES=0.6041, NP=0.022); citrate cycle (ES=0.4632, NP=0.036). The negatively correlated pathways(Fig 8D) are: natural killer cell-mediated cytotoxicity (ES=−0.5098, NP=0.018); PPAR signaling pathway (ES=−0.5307, NP=0.008); inositol phosphate metabolism (ES=−0.4302, NP=0.031).

Regulatory network of the *CXCL13* gene: This gene had novel associations with porphyrin metabolism (Porphyrin and Chlorophyll Metabolism, ES=0.5627, NP=0.071) and ascorbic acid metabolism (ES=0.6774, NP=0.057) (Fig 8E), and its negatively regulated pathways encompass: positively correlated pathways: porphyrin and chlorophyll metabolism (ES=0.5627, NP=0.071); ascorbic acid metabolism (ES=0.6774, NP=0.057); p53 signaling pathway (ES=0.4411, NP=0.066); histidine metabolism (ES=0.5000, NP=0.073); pentose and glucuronate interconversion (ES=0.6653,

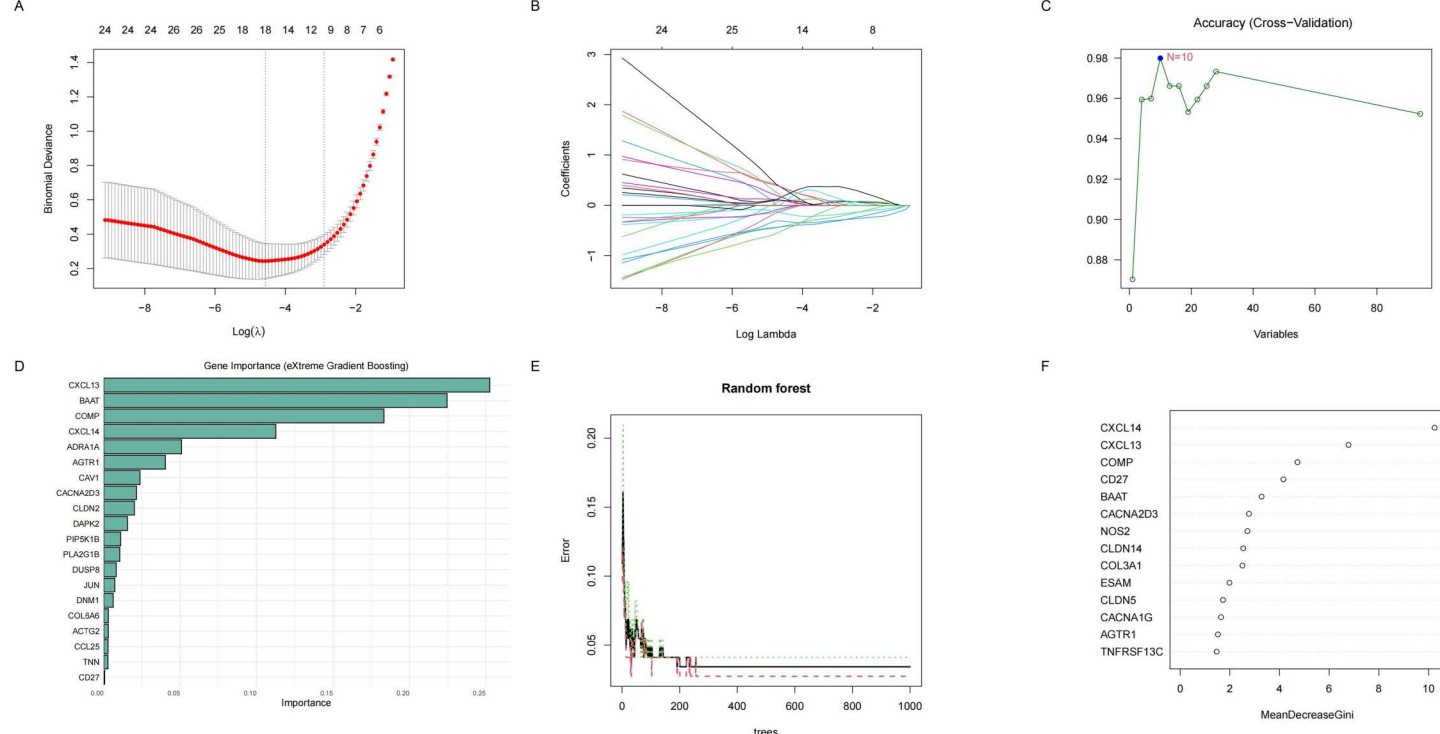

**Fig 5. Machine learning-based identification of differentially expressed genes. (A,B)** LASSO-selected genes (n = 18); **(C)** SVM-RFE-selected genes (n = 10); **(D)** XGBoost-ranked top 20 genes; **(E,F)** Random Forest-ranked top 20 genes.

NP = 0.028). Negatively correlated pathways(Fig 8F): nitrogen metabolism (ES = −0.5870, NP = 0.062); natural killer cell-mediated cytotoxicity (ES = −0.4417, NP = 0.097); vascular smooth muscle contraction (ES = −0.4069, NP = 0.051).

### 3.7 Immune cell infiltration of IPF

The infiltration of immune cells in lung tissue was analyzed(Fig 9A). In the healthy control group, immune cell composition in samples with elevated gene expression included dendritic cells (DCs), B cells, monocytes, macrophages, natural killer (NK) cells, neutrophils, CD4 + T cells, CD8 + T cells, NKT cells, and gamma delta T cells. Compared to the healthy group, the IPF group exhibited significantly increased expression of DCs, B cells, and CD4 + T cells, while the expression of other immune cells such as monocytes, neutrophils, NKT cells, and gamma delta T cells was significantly decreased(Fig 9B). The correlations among various immune cell types were analyzed using (Fig 9C).

A correlation analysis was conducted for the *BAAT*, *COMP*, and *CXCL13* genes with respect to immune cells. The results demonstrated that in high-expression cells, these three genes exhibited a significant positive correlation with B cells and CD4 + T cells, while showing a significant negative correlation with monocytes and neutrophils (Fig 10A). Subsequently, pairwise correlation analyses between each of the three genes and neutrophils, monocytes, CD4 + T cells, and B cells were performed using linear scatter plots, yielding consistent findings (Fig 10B,C,D).

### 3.8 Queue verification

In the GSE70866 database, two distinct data platforms are available. To evaluate the expression levels of three key target genes, we analyzed a dataset comprising 20 normal samples and 64 IPF samples using one of these platforms. The

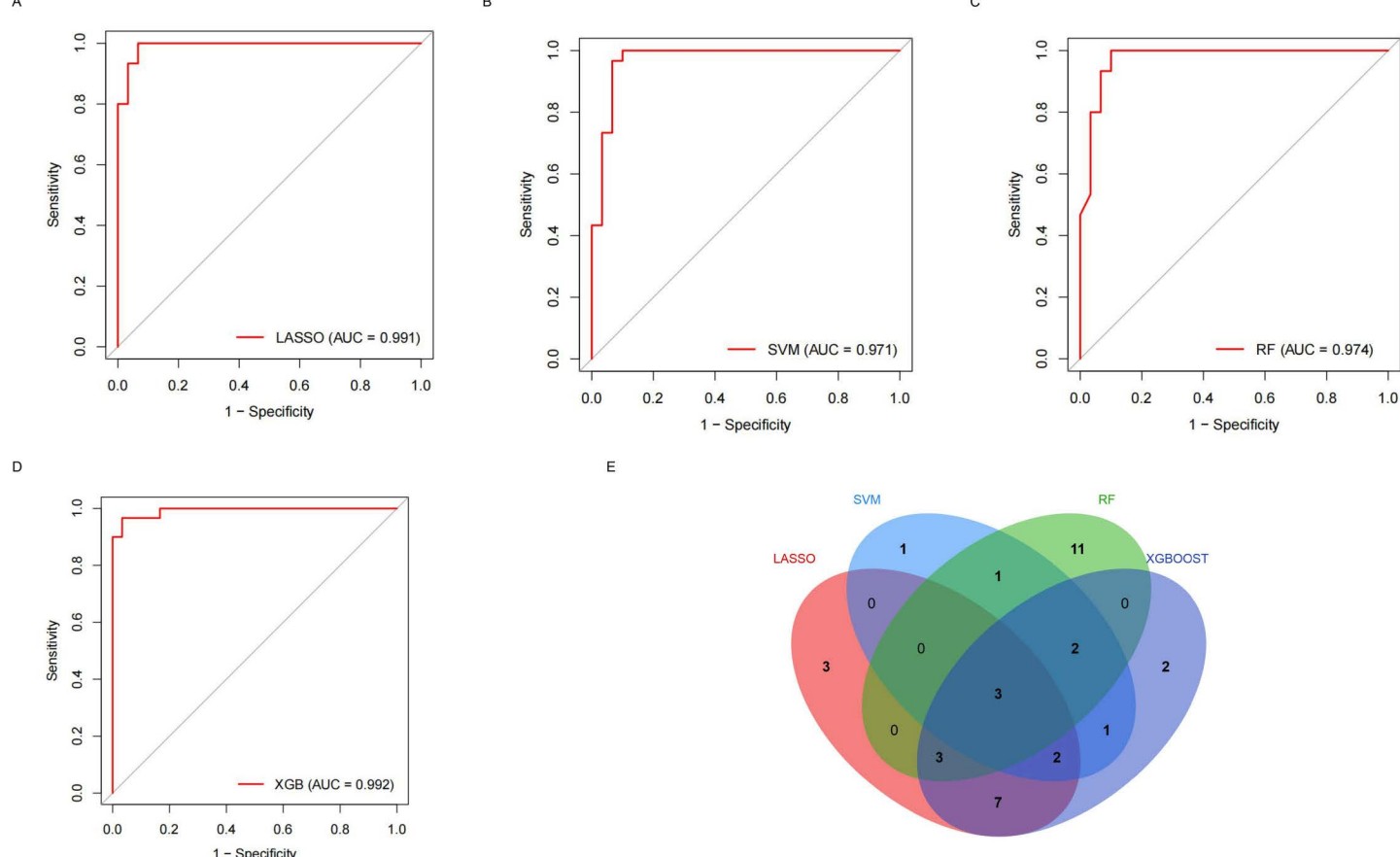

**Fig 6. Machine learning algorithm performance evaluated by ROC curves. (A)** LASSO; **(B)** SVM-RFE; **(C)** Random Forest (RF); **(D)** XGBoost; **(E)** Target genes identified through multi-algorithm validation.

results demonstrated that the IPF group exhibited significantly higher expression levels of these target genes compared to the control group (Fig 11A). Additionally, ROC curve analysis revealed that these target genes had robust predictive performance (Fig 11B).

In the GSE70866 database, the second dataset comprises 64 IPF samples. Based on the expression profiles of three hub genes, these samples were stratified into two clusters for analysis (Fig 12A,B). Differential and functional analyses between these clusters provided deeper insights into the roles of these hub genes in IPF progression (Fig 12C). Specifically, *BAAT* and *CXCL13* were highly expressed in cluster 1, while *COMP* was predominantly expressed in cluster 2 (Fig 12C). Differential gene expression analysis revealed that, compared to cluster 1, cluster 2 featured 717 upregulated genes and 191 downregulated genes (Fig 12D), indicating that most genes in this IPF subtype exhibited positive correlations with *COMP* and negative correlations with *BAAT* and *CXCL13*.

Further functional enrichment analysis was conducted on the differentially expressed genes. GO analysis revealed that these genes were significantly associated with intrinsic components of the membrane, integral components of the membrane, signaling receptor activity, molecular transducer activity, nervous system processes, and G protein-coupled receptor signaling pathways (Fig 13A). KEGG analysis indicated that the differentially expressed genes were closely linked to olfactory transduction, neuroactive ligand-receptor interaction, cAMP signaling pathways, calcium

                                                          

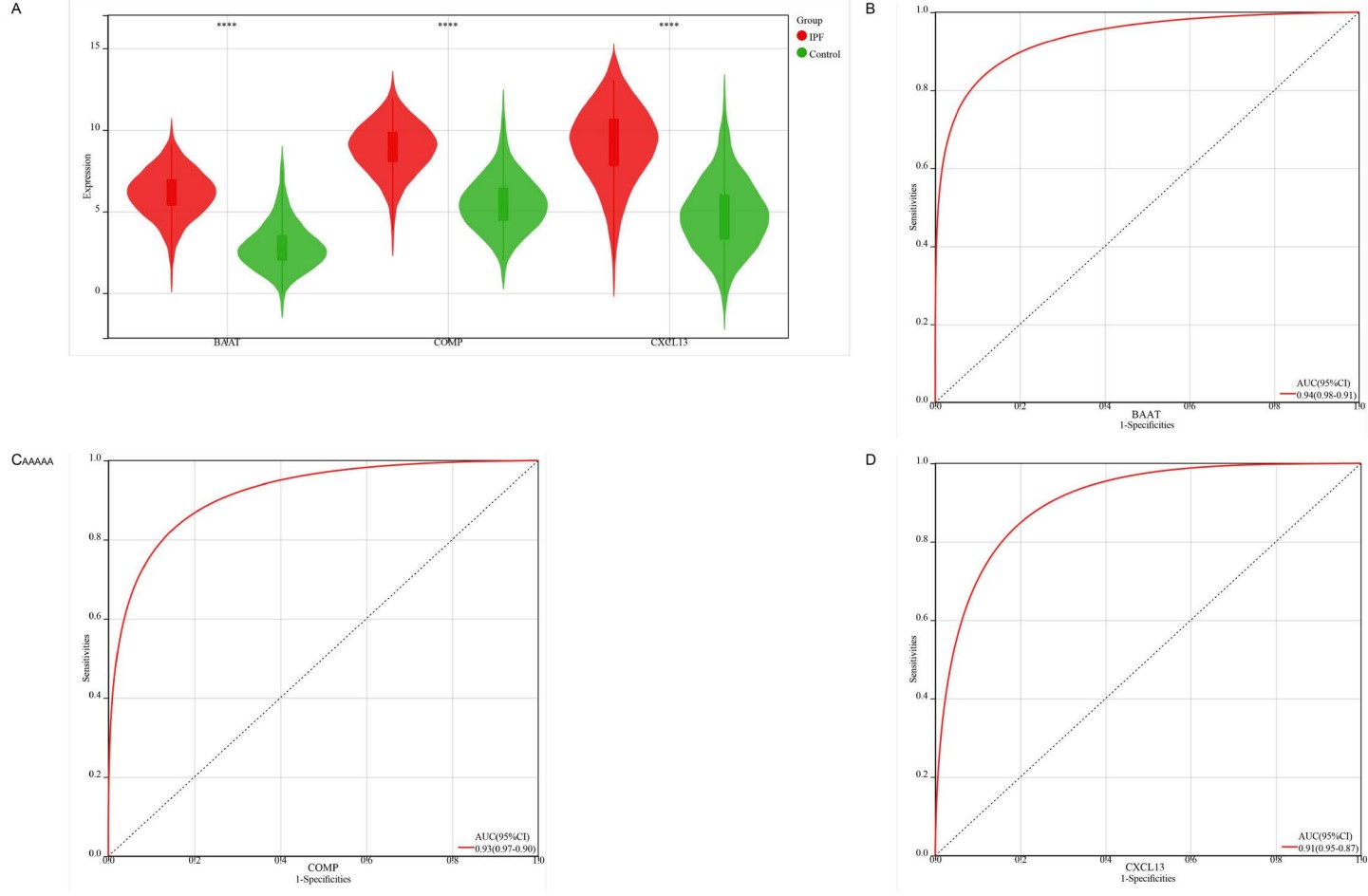

**Fig 7. Expression and diagnostic performance of the three hub genes. (A)** Violin plots showing elevated expression of BAAT, COMP, and CXCL13 in IPF samples compared to controls. **(B-D)** ROC curve analysis for BAAT **(B)**, COMP **(C)**, and CXCL13 (D) in discriminating IPF from control samples.

signaling pathways, and cGMP-PKG signaling pathways (Fig 13B). These findings support the hypothesis that the three target genes may influence the development of IPF by regulating extracellular matrix components and modulating intercellular signaling.

## 4 Discussion

IPF is a progressive lung disorder of unknown origin. Research suggests its pathogenesis involves synergistic interactions of pathological processes, particularly ECM remodeling [30] and immune cell infiltration as key contributors. Using bioinformatics methods, this study enhances understanding of molecular mechanisms underlying disease progression and provides a framework for biomarker identification.

In this study, we retrieved a GSE dataset from the GEO database to identify differentially expressed genes (DEGs) between IPF and normal lung tissues. Subsequently, GO and KEGG analyses were conducted to elucidate the biological functions of these DEGs in IPF. Compared to the control group, we identified 867 upregulated genes and 672 downregulated genes in IPF samples. Through GO and KEGG enrichment analysis, we found that these DEGs were primarily involved

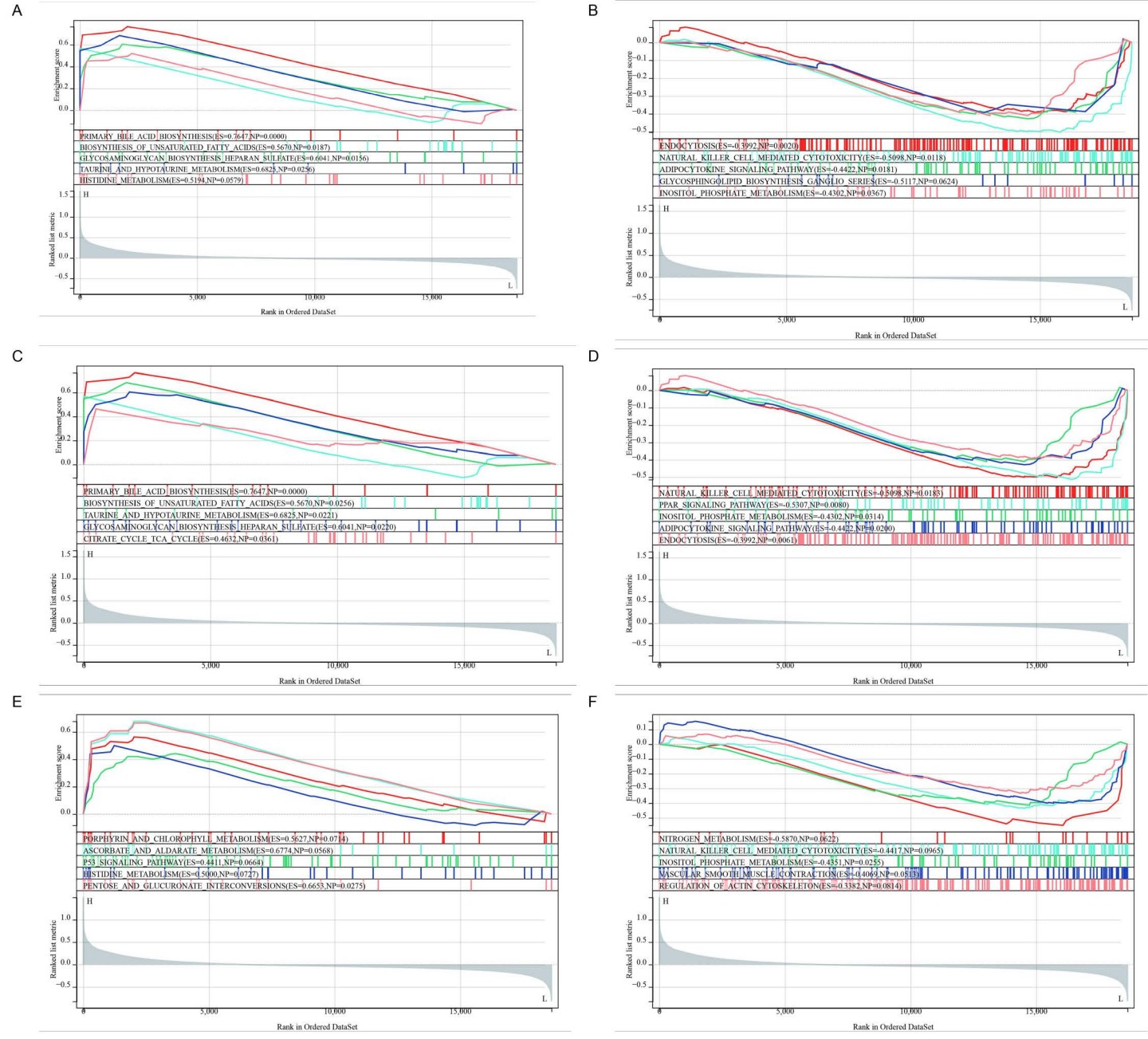

**Fig 8. Functional evaluation of three IPF-associated genes through single-gene GSEA. (A-F)** Top five correlated pathways: BAAT (A: positive, B: negative), COMP (C: positive, D: negative), CXCL13 (E: positive, F: negative).

in collagen-containing ECM, cell membrane components, and pathways such as *neuroactive ligand-receptor interaction*, *cytokine-cytokine receptor interaction*, *ECM receptor interactions*, and *renin secretion*. Functional enrichment analysis using GO and KEGG revealed that the differentially expressed genes were significantly enriched in pathways associated with collagen-containing ECM and *ECM receptor interactions.* These findings are consistent with the well-established pathological

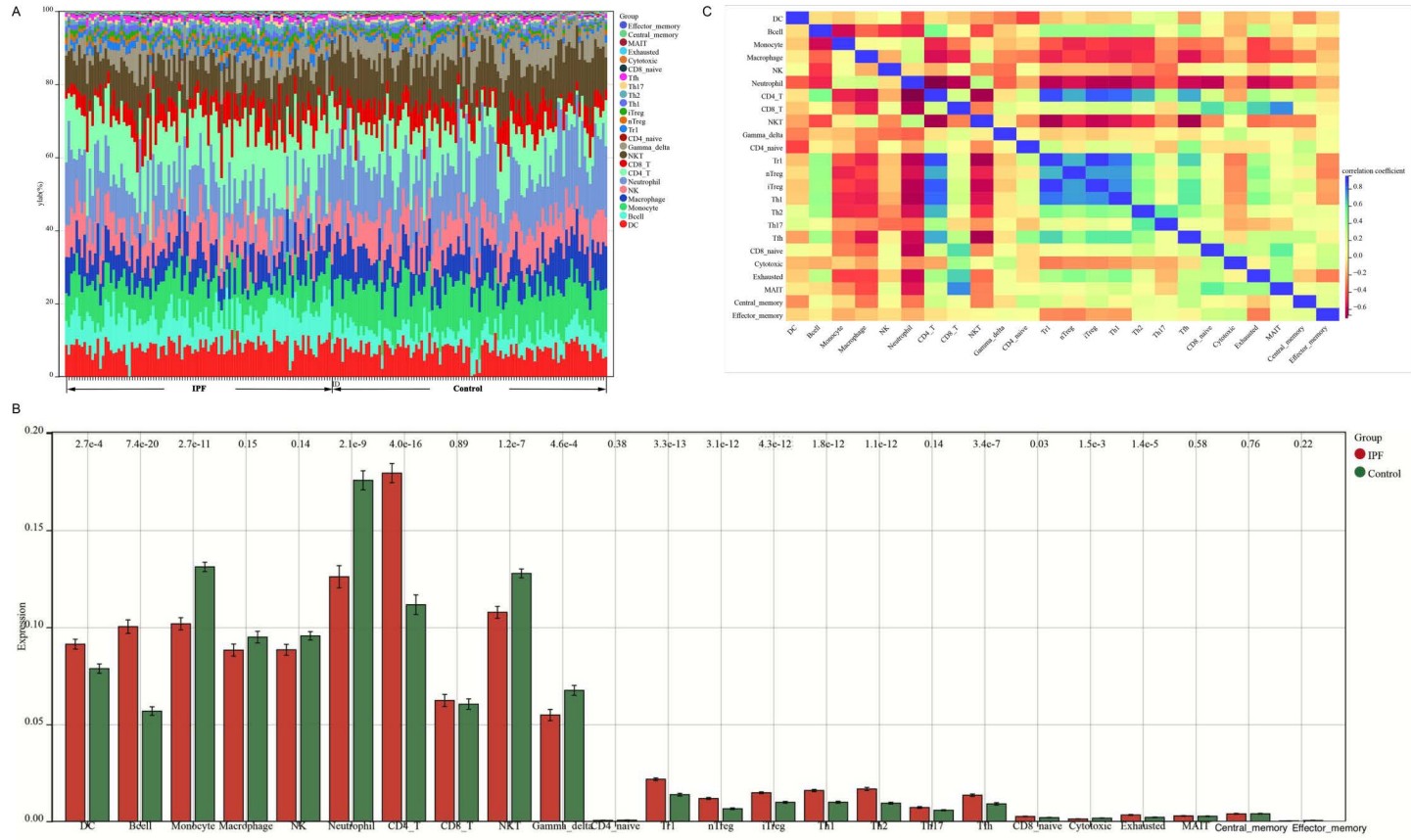

**Fig 9. Immune infiltration landscape in IPF. (A)** Lung tissue infiltration profile; **(B)** Comparative analysis: IPF vs healthy controls; **(C)** Intercellular correlation network.

hallmarks of IPF, including excessive ECM deposition and fibroblast activation, which have been corroborated by previous studies [31–33]. This consistency further substantiates the reliability and robustness of our data.

We conducted a WGCNA to identify co-expressed genes associated with IPF. This analysis generated 18 distinct co-expression modules. We selected the four modules most relevant to IPF (darkturquoise, darkgreen, darkgray, and greenyellow) for further investigation, thereby identifying 1,423 genes that were highly correlated with IPF. Subsequently, we performed an intersection analysis of 1,539 DEGs, the 1,423 co-expression hub genes obtained through WGCNA, and 1,193 ERGs retrieved from the GSEA database. This analysis revealed 94 ERDEGs associated with IPF. Nance et al. [34], through transcriptome analysis of alternative splicing events in IPF fibroblasts, identified a significant number of IPF differential genes. Notably, the aberrant expression of *COL1A1*, *FN1* (fibronectin), and *TGFB1* was closely linked to ECM remodeling. These studies provided a theoretical foundation for our identification of target genes related to ECM, differential genes, and IPF. Additionally, functional enrichment analysis confirmed the strong association between these ERDEGs and the ECM.

In this study, we employed a suite of rigorously cross-validated machine learning techniques—LASSO, SVM-RFE, RF, and XGBoost—to identify key feature genes among differentially expressed ECM genes associated with IPF. By integrating the gene rankings derived from these four algorithms, we identified three core genes: *BAAT*, *COMP*, and *CXCL13*. Quantitative analysis revealed that the expression levels of these genes were significantly elevated in IPF lung tissues

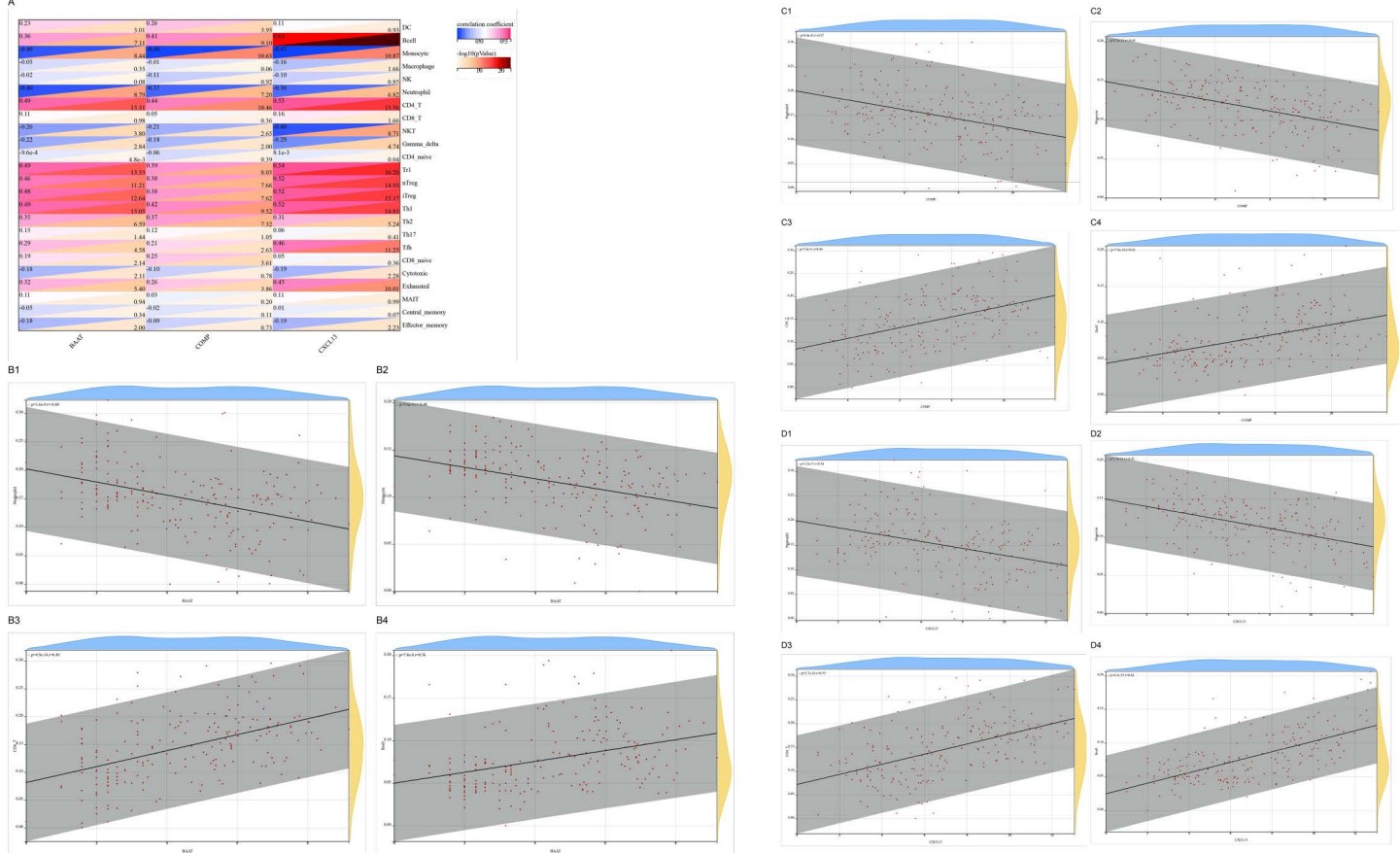

**Fig 10. Immune correlation analysis of three candidate genes. (A)** Pan-gene correlation; **(B-D)** Gene-specific associations: BAAT **(B)**, COMP **(C)**, CXCL13 (D) with neutrophils, monocytes, CD4+T cells, and B cells (linear regression).

compared to healthy controls (p < 0.01 after FDR correction), with area under the curve (AUC) values ranging from 0.85 to 0.92, indicating robust disease discrimination capability. Our approach aligns with recent advancements in the field. For instance, Zhang Y et al. [35] utilized a LASSO-SVM-RFE combination to screen for candidate biomarkers of IPF and explore their correlations with immune cells, as well as their potential as immunotherapeutic targets for IPF. Meanwhile, Wu Z et al. [36] identified IPF biomarkers using multiple machine learning methods and evaluated the role of immune infiltration in the disease, highlighting the synergistic advantages of combining various machine learning approaches for data integration. These methodological agreements have significantly improved the precision of biomarker identification.

Following single-gene analyses of *BAAT*, *COMP*, and *CXCL13*, we observed that the expression levels of all three genes were significantly elevated in the IPF group, demonstrating their potential as robust diagnostic markers. The bile acid-CoA:amino acid N-acyltransferase, encoded by the *BAAT* gene, catalyzes the conjugation of primary bile acids—such as cholic acid and chenodeoxycholic acid—with glycine or taurine via amide bonds in the liver, resulting in the formation of conjugated bile acids. This biochemical process represents a key step in bile acid metabolism. Historically, the liver has been considered the primary site of *BAAT* activity, with bile acids detected in lung tissue believed to originate either from hepatic synthesis followed by systemic circulation or to be indirectly regulated by gut microbiota through the gut-lung axis. However, accumulating evidence suggests that under pathological conditions, *BAAT* expression and its potential functions in the lungs have increasingly attracted scientific attention.Chen et al. [37] reported activation of the bile acid

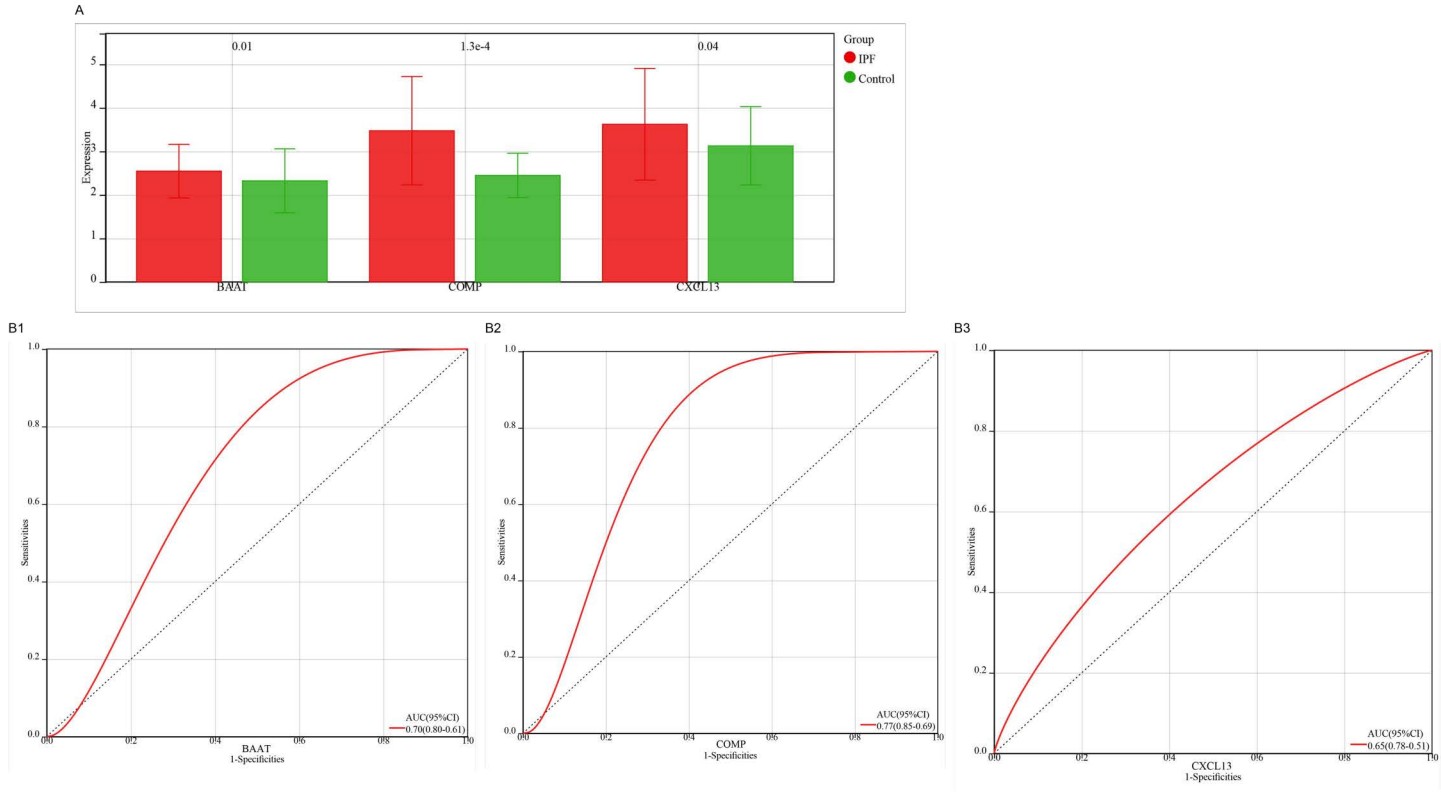

**Fig 11. Expression validation of three genes using GSE70866 dataset. (A)** Differential expression: IPF vs controls; **(B)** Diagnostic performance: BAAT, COMP, CXCL13 (ROC analysis).

signaling pathway in the lung tissue of individuals with pulmonary fibrosis, primarily mediated through the FXR receptor, suggesting the possibility of localized bile acid metabolic disturbances. This hypothesis is further supported by single-cell transcriptomic data. Public databases, such as the Human Protein Atlas, indicate that *BAAT* mRNA expression is minimal in healthy lung tissue but significantly elevated in patients with IPF. Adams TS et al. [38] further demonstrated that *BAAT* expression in IPF lungs exhibits distinct cellular specificity, with transcriptional signals predominantly localized in aberrant basal-like cell populations. Histopathological analysis reveals that these cells are specifically located at the leading edges of myofibroblast foci and display unique molecular features, including co-expression of epithelial markers (KRT5 and TP63), basal cell markers (NGFR and ITGA6), epithelial-mesenchymal transition (EMT)-related genes (VIM and CDH2), and senescence-associated genes (CDKN2A). Notably, these cells also exhibit high expression of the transcription factor SOX9, which has been identified as a key regulator of distal airway development and tissue repair.Moreover, Habermann AC et al. [39] observed that *BAAT* is specifically enriched in KRT5−/KRT17+cells—a pathological cell population associated with excessive ECM production. This phenotype is driven by activation of the SOX4/SOX9 pathway and suppression of the NR1D1 signaling network. RNA in situ hybridization (RNA-ISH) results further confirm significant co-localization of KRT17 and SOX9 in fibrotic lung tissues, providing direct evidence for the involvement of dysregulated epithelial cells in collagen synthesis.In summary, *BAAT* is markedly upregulated in the lungs of IPF patients, and its cellular origin is closely associated with pathological airway remodeling and abnormal ECM accumulation. We propose that the enzyme encoded by *BAAT* may exhibit catalytic activity in the lung and contribute to pulmonary fibrosis through several potential mechanisms:1. Local bile acid signaling pathway [37]. Conjugated bile acids generated by *BAAT* may activate alveolar epithelial

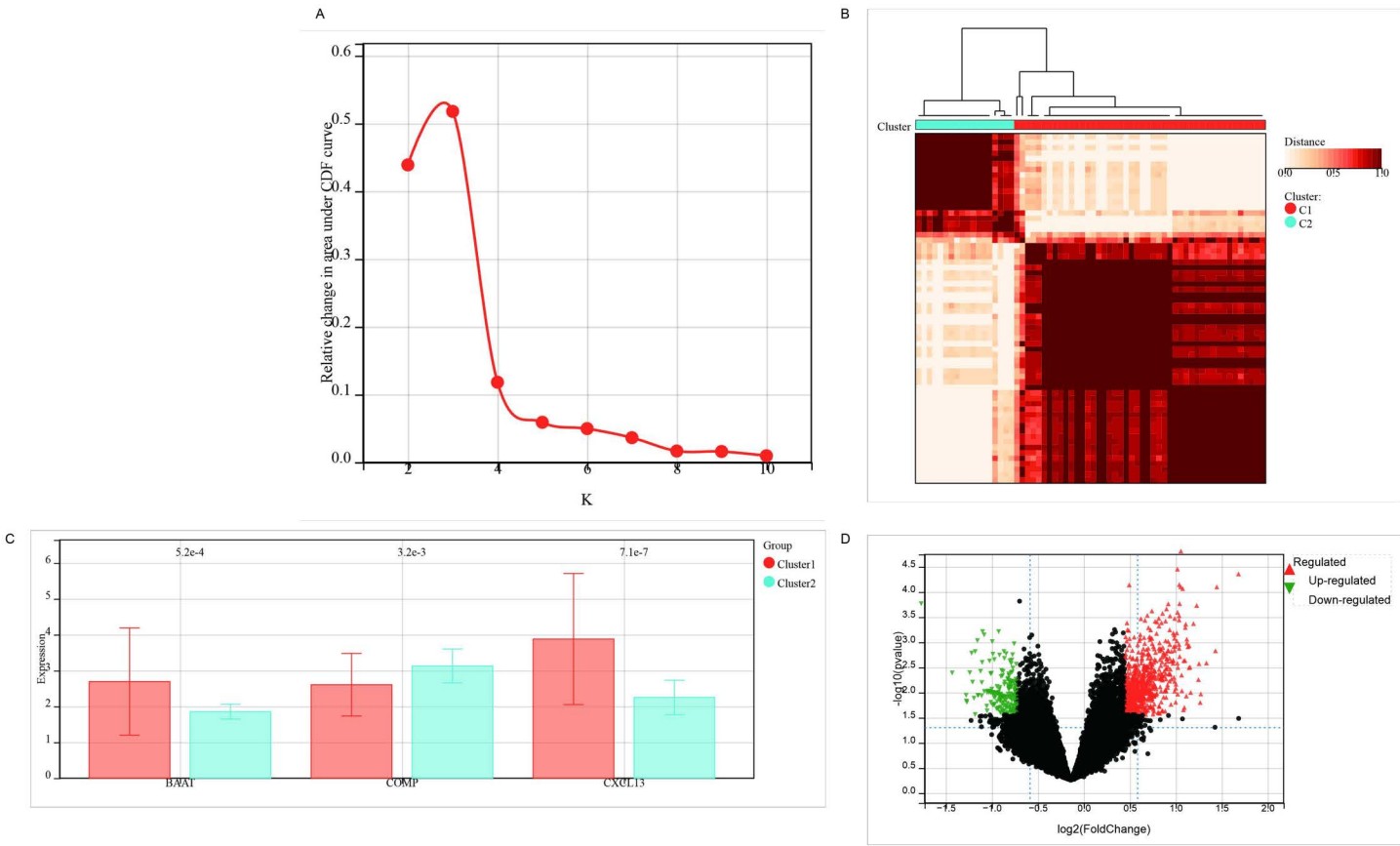

**Fig 12. Molecular subtyping of ILD through tri-gene clustering. (A,B)** Consensus clustering; **(C)** Subgroup expression patterns; **(D)** Inter-subgroup differential expression.

cells or fibroblasts via FXR or TGR5 receptors, thereby promoting the expression and release of pro-fibrotic mediators such as TGF-β and CTGF.2. Immune-metabolic interaction pathway [40]: Conjugated bile acids may modulate the function of lung immune cells, including Th17 cells and macrophages, thereby indirectly promoting ECM deposition.3. Oxidative stress and apoptosis pathway [41]: Studies have shown that conjugated bile acids accumulate in T cells of liver cancer patients, impairing mitochondrial respiration and inducing reactive oxygen species (ROS) production. We speculate that similar pathophysiological changes may occur in the lungs of IPF patients, where bile acid metabolic imbalance may lead to ROS generation, subsequently activating pro-fibrotic enzymes such as lysyl oxidase (LOX), exacerbating collagen cross-linking (in synergy with MMP/TIMP imbalance), and inducing apoptosis.Despite these insights, significant gaps remain in understanding the precise role of BAAT in IPF. First, the conjugation of bile acids requires the activation of primary bile acids and coenzyme A (CoA), yet the lung lacks the complete bile acid synthesis enzyme system (e.g., CYP7A1, CYP27A1). Whether the lung can sustain local bile acid metabolism through the uptake of circulating primary bile acids remains to be determined. Second, although *BAAT* is expressed in the lung, its baseline levels are very low, raising doubts about whether the resulting conjugated bile acids are sufficient to exert meaningful physiological effects.Therefore, future studies should focus on clarifying the functional status of *BAAT* in lung tissue: examining *BAAT* protein expression via immunohistochemistry or Western blotting; evaluating its enzymatic activity using lung tissue homogenates; and comparing the ratios of conjugated to free bile acids in IPF and control lung tissues using metabolomic approaches. Given

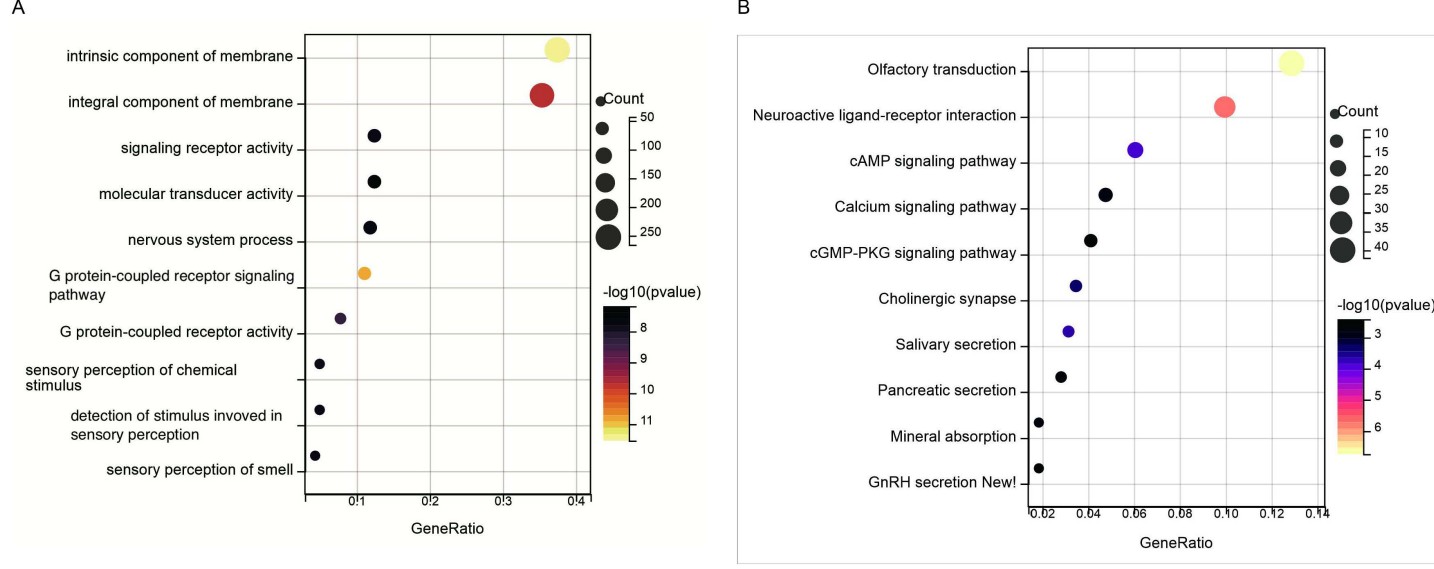

**Fig 13. Functional enrichment of DEGs. (A)** GO analysis; **(B)** KEGG pathway analysis.

the low concentrations of bile acids in lung tissue, such analyses will likely require highly sensitive techniques, such as liquid chromatography-mass spectrometry (LC-MS).In conclusion, although the upregulation of *BAAT* in the lungs of IPF patients has been documented, there is currently no direct evidence confirming its catalytic activity in this organ. If such activity is validated, it could open new avenues for understanding the metabolic-immune regulatory mechanisms in IPF, offering both theoretical and clinical implications.

The *COMP* gene encodes cartilage oligomeric matrix protein (COMP), a non-collagenous glycoprotein belonging to the thrombospondin family and functioning as a key component of ECM. Single-cell RNA sequencing (scRNA-seq) data demonstrate that *COMP* is predominantly expressed in myofibroblasts and fibroblasts and represents one of the most significantly upregulated genes in lung tissues from patients with IPF. Compared to healthy controls, COMP expression levels are reported to increase by approximately 17-fold [38], a result that has been validated at the protein level using Western blot analysis. Immunohistochemical staining further reveals that *COMP* is highly localized in the dense fibrotic regions of IPF lungs, where it co-localizes with vimentin and is spatially associated with cells expressing phosphorylated SMAD3 (pSMAD3). ELISA measurements also show that serum COMP levels are significantly elevated in IPF patients compared to healthy individuals.From a pathogenic perspective, IPF is characterized by excessive fibroblast proliferation, enhanced differentiation into myofibroblasts, and excessive deposition of collagen. *COMP*'s marked overexpression in these cell types underscores its critical involvement in the fibrotic process at the cellular level. These observations are consistent with findings from studies by Adams TS et al. [38] and Habermann AC et al. [39]. Further mechanistic insights were provided by Vuga LJ et al. [42], who demonstrated that *COMP* promotes fibroblast activation through upregulation of TGF-β1 and enhanced SMAD3 phosphorylation *(pSMAD3↑)*, thereby facilitating their transformation into myofibroblasts. This leads to increased expression of *α-SMA* and *COL1A1* and suppression of MMP activity, ultimately reducing ECM degradation. Experimental knockdown of *COMP* has been shown to significantly inhibit the expression of these fibrotic markers in pulmonary fibroblasts. Moreover, *COMP* plays a central role in mediating pro-fibrotic crosstalk between macrophages and fibroblasts.

*CXCL13* is a member of the CXC chemokine family, primarily secreted by stromal cells in lymphoid tissues. It specifically binds to the CXCR5 receptor, guiding the migration of B cells and follicular helper T cells (Tfh) into lymphoid

follicles [43]. In the context of IPF, *CXCL13* promotes the aggregation of B and T cells within lung tissue, contributing to the formation of ectopic lymphoid follicles. These structures are implicated in antigen presentation, autoantibody production, and the organization of TLS [44]. Research by Nessrine Bellamri et al. [45] identified CD68+ and CD206+alveolar macrophages as the primary cellular source of *CXCL13* in IPF patients. This finding was further supported by Morse et al. [46], who used single-cell RNA sequencing to show that *CXCL13* is significantly upregulated in abnormal basal cell populations, providing strong evidence for its cellular origin.Asai Y et al. [47] demonstrated that a subset of CD4+T cells, specifically Tfh cells, expresses the CXCR5 receptor. Upon binding to *CXCL13*, this interaction facilitates the migration of Tfh cells into B cell follicles. These Tfh cells express co-stimulatory molecules such as ICOS and immune regulatory molecules like PD-1, and utilize BCL6 as a master transcription factor to secrete cytokines including IL-4, IL-10, and IL-21, which are essential for B cell activation, differentiation, and class switching. Excessive activation of Tfh cells may lead to antigen-specific antibody responses and potentially contribute to autoimmune phenomena. Notably, Morse et al. [46] also observed significant alterations in the CD4+Treg subset (FOXP3+) in IPF lung tissues compared to controls, suggesting a potential role for regulatory T cells in disease pathogenesis.In addition to its immunomodulatory functions, CXCL13 can activate lung fibroblasts via the CXCR5 receptor, leading to upregulation of α-SMA and type I collagen expression and thereby promoting extracellular matrix accumulation. CXCL13 also synergizes with TGF-β to enhance its pro-fibrotic effects through increased SMAD3 phosphorylation. Recent clinical studies have shown that serum CXCL13 levels are significantly elevated in IPF patients and are positively correlated with disease severity and progression rate [48].

A systematic analysis of immune cell infiltration in lung tissue revealed that the genes *BAAT*, *COMP,* and *CXCL13* were significantly positively correlated with B cells and CD4+T cells but negatively correlated with monocytes and neutrophils. Multiple single-cell sequencing studies have confirmed significant dysregulation in the composition of immune cells in the lung tissue of IPF patients. Misharin et al. [49] demonstrated that in healthy lung tissue, alveolar macrophages (comprising 60%−70% of CD45+cells) and resident memory T cells form the foundation of immune homeostasis. In contrast, IPF patients exhibit the following characteristic alterations in immune cell profiles:

1. Abnormal activation of dendritic cells (DCs): In IPF patients, the CD1c+ conventional dendritic cell subset is significantly expanded. These cells highly express *MHC-II* and co-stimulatory molecules (such as CD80/CD86), potentially activating specific T cell responses by presenting ECM degradation products like *COMP*. Posey KL et al. [50] further showed that *COMP* enhances *TGF-β* activity, which in turn increases *COMP* expression, forming a positive feedback loop. This leads to increased ECM synthesis, perpetuating the fibrotic cycle, and promoting type I collagen expression, collagen fiber reorganization, and ECM stiffening.

2. Clonal proliferation of B cells: Ruffmann et al. [51] reported IgG+ plasma cell infiltration and TLS formation in the lung tissues of IPF patients. Cosgrove J et al. [52] found that *CXCL13* is a key chemokine for TLS formation, with high expression levels and a significant positive correlation with B cell aggregation (r=0.78, p<0.001). This suggests that *BAAT*, *COMP*, and *CXCL13* may be involved in autoantibody generation through regulation of B cell receptor signaling pathways such as *BLyS/BAFF*.

3. *Th2/Th17* polarization: In IPF patients, CD4+T cells tend to secrete IL-4/IL-13 (Th2) and IL-17A (Th17). Spagnolo P et al. [53] indicated that Th2 cells promote fibroblast transformation via *STAT6* pathway activation. Han et al. [40] found that this phenomenon might be related to *BAAT*'s bile acid metabolic function, where bile acid metabolites (such as deoxycholic acid) enhance Th17 cell differentiation.

4. Functional shift in the monocyte-macrophage axis: Despite reduced monocyte numbers, alveolar macrophages (AMs) in IPF patients exhibit a pro-fibrotic phenotype (M2 polarization). *COMP* enhances macrophage adhesion through integrin αVβ3, explaining its negative correlation with monocytes. Chrysanthopoulou et al. [54] noted that neutrophil extracellular traps (NETs) released by neutrophils in IPF patients can activate fibroblasts via TLR9. The reduction in

neutrophil infiltration may be stage-dependent, with neutrophils predominating in early stages but being replaced by other cells later, accounting for the negative correlations between *BAAT*, *COMP*, and *CXCL13* and neutrophils in IPF patients.

5. Loss of protective effects of NKT cells and γδ T cells: Akbari et al. [55] found that in healthy lungs, NKT cells secrete IL-10 to suppress excessive inflammation. Reduced NKT cell counts in IPF may lead to immune suppression imbalance and accelerate fibrosis. The negative correlation with *BAAT* and *CXCL13* indicates an interaction between metabolism and immunity.

The above results demonstrate that the three target genes are closely associated with immune cell infiltration in IPF. Furthermore, their mechanisms of action are complex and influence multiple signaling pathways, thereby providing a critical foundation for future research.

Finally, we conducted cross-validation and clustering analysis on the three genes, and the results were consistent with the previous differential gene and functional enrichment analysis, further confirming that the target genes may affect the occurrence and development of IPF by regulating extracellular matrix components and influencing intercellular signal transduction.

Based on the pathological mechanism of IPF, this research integrated differential gene expression, ECM remodeling, and immune factors to explore the application value of non-invasive gene analysis in this disorder and identified three ECM-related genes (*BAAT*, *COMP*, and *CXCL13*) highly correlated with IPF. From the aspect of pathological mechanism, the injury repair process in IPF patients is dysregulated, resulting in abnormal activation of fibroblasts and excessive deposition of ECM. ECM remodeling is not merely the outcome of fibrosis but also actively promotes the fibrotic process by enhancing tissue stiffness and signal transduction. Additionally, the infiltration of immune cells (such as macrophages and T cells) releases pro-fibrotic factors, forming a vicious cycle and further aggravating disease progression. Non-invasive gene analysis, encompassing biomarker detection in blood or bronchoalveolar lavage fluid (BALF), such as the gene combination discovered in this study, might offer novel tools for early diagnosis and disease monitoring. Specifically, *COMP*, as an ECM component, has been extensively investigated, *CXCL13* is closely associated with immune cell infiltration, and *BAAT* may be implicated in metabolic abnormalities, providing a fresh perspective for understanding the pathogenesis of IPF.

## 5 Deficiencies and further perspectives

This study, as a previous research project of our research group, primarily concentrates on bioinformatics analysis, with the aim of offering a theoretical foundation for subsequent experiments. Through bioinformatics analysis, we have initially identified a batch of potential target genes, which might play significant roles in specific biological processes. Nevertheless, as the current research is confined to the data analysis aspect and lacks experimental validation, the content is relatively tenuous and has failed to fully disclose the specific functions of these genes and their action mechanisms in the occurrence and development of diseases.

To further deepen the research, the following aspects of work will be the focus of our next steps: 1. Experimental verification of target genes: Based on the target genes screened out in the previous stage, we will undertake expression level and protein level verifications through various experimental approaches (such as qPCR, Western Blot, immunofluorescence, etc.) to confirm the actual expressions of these genes in cell or tissue samples. This step is of vital importance for ensuring the reliability and accuracy of the subsequent research. 2. Target gene knockout experiments: Utilizing gene editing techniques like CRISPR/Cas9, we will carry out knockout treatments on selected target genes and observe their influences on cell phenotypes, signaling pathways, and biological functions. By establishing stable gene knockout cell lines, we can explore the functions of these genes more profoundly and lay the groundwork for further mechanism research.

3. In vivo experimental verification: After obtaining preliminary results from in vitro experiments, we will further conduct animal model experiments, such as mouse models, to verify the functions of target genes in vivo and their correlations with diseases. By establishing appropriate disease models, we can better simulate the pathological processes of human diseases, thereby providing valuable references for clinical applications.

In summary, although this study has only accomplished the bioinformatics part at present, through a series of rigorous experimental validations and the application of technical means in the future, we are confident that we can gradually uncover the functions of target genes and their action mechanisms in related diseases, providing solid theoretical and technical support for future clinical translational research.

IPF is a chronic, progressive interstitial lung disease with an unknown etiology. Its pathological features are marked by significant cellular heterogeneity and complex spatially organized tissue dynamics. Conventional bulk RNA sequencing approaches are limited in their ability to resolve the intricate cellular composition and spatial organization within the IPF lung. In contrast, the integration of scRNA-seq and spatial transcriptomic technologies (e.g., Visium and Xenium) offers a powerful strategy for dissecting cellular heterogeneity at high resolution. These technologies enable precise mapping of cell localization within fibrotic lesions (such as fibroblast foci) or inflammatory infiltrates, and facilitate the elucidation of intercellular communication networks, particularly the mechanisms by which immune cells (e.g., macrophages and T cells) activate fibroblasts via ligand-receptor interactions. Furthermore, the application of computational tools such as CellPhoneDB and NicheNet allows for the prediction of spatially coordinated signaling pathways, thereby enabling a more comprehensive understanding of the dynamic shifts in cellular populations and molecular profiles during disease progression from early inflammatory to late fibrotic stages. The anticipated outcomes include the construction of a high-resolution cellular atlas of IPF lung tissue through the analysis of disease-specific core genes, the identification of distinct fibroblast subpopulations with pro-fibrotic functions, and the discovery of novel immune cell subsets involved in the regulation of fibrotic processes. In terms of spatial organization and microenvironmental interactions, co-localization analyses of gene expression will provide insights into the functional mechanisms underlying pro-fibrotic pathways. Through target identification and functional validation, this study aims to uncover novel therapeutic candidates for the treatment of IPF. The integration of single-cell and spatial multi-omics data will transition IPF research from traditional, tissue-level "black box" analyses to a precise, integrative framework encompassing "cell-molecule-space", thereby offering a robust theoretical foundation and technological platform for the development of novel anti-fibrotic therapies. Additionally, this study will investigate the molecular mechanisms of key genes such as BAAT, COMP, and CXCL13, and further assess whether these molecules are co-expressed within the same cellular subpopulations (e.g., matrix-immune hybrid populations) and whether they exhibit spatial co-localization, potentially contributing to the formation of a pro-fibrotic microenvironment.

## Acknowledgments

We gratefully acknowledge the BioBean Bioinformatics Platform (http://www.sxdyc.com/, accessed 15 November 2024) for computational infrastructure support. We also thank Dr. X for statistical consultation.

## Author contributions

**Data curation:** Man Wang.

**Formal analysis:** Man Wang, Lu Liu.

**Investigation:** Lu Liu, Yang Liu.

**Writing – original draft:** Man Wang.

**Writing – review & editing:** Shihuan Yu.

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
