## [Decision Letter · Decision Letter 0]

28 Jul 2025

Dear Dr. Yu,

Thank you for submitting your manuscript to PLOS ONE. After careful consideration, we feel that it has merit but does not fully meet PLOS ONE’s publication criteria as it currently stands. Therefore, we invite you to submit a revised version of the manuscript that addresses the points raised during the review process.

**ACADEMIC EDITOR:**

Please respond to the reviewers' comments by taking the necessary steps.

We look forward to receiving your revised manuscript.

Kind regards,

Yoshiaki Zaizen, MD, PhD

Academic Editor

PLOS ONE

Journal Requirements:

https://www.nature.com/articles/s41598-023-43834-z

In your revision ensure you cite all your sources (including your own works), and quote or rephrase any duplicated text outside the methods section. Further consideration is dependent on these concerns being addressed.

6. Please amend the manuscript submission data (via Edit Submission) to include author YangLiu.

7. Please upload a new copy of Figures 2, 10 as the detail is not clear. Please follow the link for more information: "https://blogs.plos.org/plos/2019/06/looking-good-tips-for-creating-your-plos-figures-graphics/" https://blogs.plos.org/plos/2019/06/looking-good-tips-for-creating-your-plos-figures-graphics/

8. Please ensure that you refer to Figure 13 in your text as, if accepted, production will need this reference to link the reader to the figure.

9. Please remove your figures from within your manuscript file, leaving only the individual TIFF/EPS image files, uploaded separately. These will be automatically included in the reviewers’ PDF.

10. Please include captions for your Supporting Information files at the end of your manuscript, and update any in-text citations to match accordingly. Please see our Supporting Information guidelines for more information: http://journals.plos.org/plosone/s/supporting-information.

Reviewers' comments:

Reviewer's Responses to Questions

**Comments to the Author**

1. Is the manuscript technically sound, and do the data support the conclusions?

Reviewer #1: Yes

Reviewer #2: Yes

2. Has the statistical analysis been performed appropriately and rigorously?

Reviewer #1: I Don't Know

Reviewer #2: Yes

3. Have the authors made all data underlying the findings in their manuscript fully available?

Reviewer #1: Yes

Reviewer #2: Yes

4. Is the manuscript presented in an intelligible fashion and written in standard English?

Reviewer #1: Yes

Reviewer #2: Yes

Reviewer #1: General Comments

This manuscript by Wang et al. presents a comprehensive bioinformatics re-analysis of publicly available datasets to identify key genes associated with Idiopathic Pulmonary Fibrosis (IPF). The study successfully identifies BAAT, COMP, and CXCL13 as significantly upregulated genes in IPF. The identification of BAAT, a gene encoding an enzyme involved in bile acid metabolism, and its potential link to the infiltration of B cells and CD4+ T cells, is particularly novel and interesting.

The primary limitation, as the authors rightly acknowledge, is the lack of experimental validation to confirm the functional roles of these genes in IPF pathogenesis. However, the study provides a solid foundation for future experimental work.

To further enhance the impact and clarity of the manuscript, I offer the following suggestions for the authors' consideration.

Major Comments

Strengthening the Biological Context by Identifying the Cellular Source of Key Genes: A crucial point for discussion is the cellular origin of BAAT, COMP, and CXCL13 in the IPF lung. While this may not be derivable from the bulk datasets used in this study, recent single-cell RNA-seq (scRNA-seq) resources, such as the IPF Cell Atlas (http://www.ipfcellatlas.com/), could provide valuable insights. For instance, analysis of scRNA-seq data indicates that BAAT is almost exclusively expressed in aberrant basaloid cells, and COMP is highly expressed in myofibroblasts—both of which are key pathogenic cell types in IPF. Integrating these findings into the discussion would significantly strengthen the authors' argument for the importance of these genes, as it directly links them to the primary cellular drivers of the disease. This would provide a more compelling mechanistic hypothesis for the roles of BAAT and COMP.

Minor Comments

1. Figure 9A (Immune Cell Infiltration):

In Figure 9A, which illustrates immune cell infiltration patterns, only the IPF group is shown. It would be highly beneficial to display the data for the Control group alongside the IPF group. This direct comparison is essential for readers to fully appreciate the differences in the immune landscape between healthy and diseased states, as analyzed in Figure 9B.

2. Clarity and Readability of Figures and Legends:

The overall presentation of the figures could be improved for better readability. Many figures are difficult to interpret due to small font sizes and cluttered layouts. I recommend increasing the font size and reorganizing the layout to make them more reader-friendly.

Furthermore, the figure legends could be revised for sufficient information with clarity and conciseness. For example, the legend for Figure 7 could be simplified to something like:

"Figure 7. Expression and diagnostic performance of the three hub genes. (A) Violin plots showing elevated expression of BAAT, COMP, and CXCL13 in IPF samples compared to controls. (B-D) ROC curve analysis for BAAT (B), COMP (C), and CXCL13 (D) in discriminating IPF from control samples."

3. Clarification of Datasets in Methods:

In the Methods section (Section 2.1), it would be clearer to explicitly state the technology platform for the datasets used. For instance, specifying that GSE150910 is an RNA-seq dataset and GSE70866 is a microarray dataset, both derived from lung tissue samples of IPF and control individuals, would provide important context for the reader.

4. Wording of Correlation Analysis Results:

On page 17, in the first sentence of the results section describing the immune correlation analysis, the phrase "in high-expression cells" seems imprecise. The correlation is between the gene expression level of a bulk tissue sample and the estimated immune cell abundance within that same sample. Therefore, "in samples with high gene expression" or "in high-expression individuals" would be more accurate. Please clarify this wording.

In conclusion, this is a valuable study that provides novel insights into the molecular landscape of IPF. I believe that addressing these points will significantly improve the manuscript. I look forward to seeing a revised version.

Reviewer #2: Comments to the Author

This study has reported novel and clinically applicable findings based on cutting-edge and accurate analytical methods, and can provide new insights into the development of biomarkers and novel treatments for IPF. Please provide comment to the query below.

Major comments

In this study, BAAT, which is related to bile acid metabolism, and CXCL13, which may induce TLS in IPF, were detected as major ECM-related genes in IPF, which are novel and pathologically significant. As the author say, the next important task is to clarify the cells expressing these genes, their localization, and their interactions with other molecules using single-cell RNA sequencing and spatial transcriptome analysis, etc. Please reveal in the discussion how you consider about importance of such analysis and the expected results. It has been reported that CXCL13 is produced by CD68, CD206 positive macrophages (Bellamri, et al. J. Immunol. 2020, 204, 2492–2502. ).

In addition, is it possible that BAAT, which is related to bile acid metabolism, induces ER stress and apoptosis through its effect on lipid metabolism? Please present your consideration on this issue, showing the limitations of your inference.

**Do you want your identity to be public for this peer review?** For information about this choice, including consent withdrawal, please see our Privacy Policy

Reviewer #1: **Yes: ** Toyoshi Yanagihara

Reviewer #2: No

---

## [Author Response · Author response to Decision Letter 1]

3 Aug 2025

We have made every effort to improve the manuscript by implementing certain revisions, which do not affect the overall content or structure of the paper. These changes have been highlighted in red in the revised version and are not listed here.

We sincerely appreciate the Editors’ and Reviewers’ dedicated and thoughtful work. We hope the revised manuscript meets your approval.

Once again, thank you very much for your valuable comments and suggestions.

---

## [Decision Letter · Decision Letter 1]

6 Aug 2025

Identification of Core Genes in the Extracellular Matrix and the Regulatory Mechanisms of the Immune Microenvironment in Idiopathic Pulmonary Fibrosis Using WGCNA and Machine Learning Methods

PONE-D-25-20083R1

Dear Dr. Yu,

We’re pleased to inform you that your manuscript has been judged scientifically suitable for publication and will be formally accepted for publication once it meets all outstanding technical requirements.

Kind regards,

Yoshiaki Zaizen, MD, PhD

Academic Editor

PLOS ONE

Additional Editor Comments (optional):

Thank you for submitting your manuscript to PLOS One.

Reviewers' comments:

Reviewer's Responses to Questions

**Comments to the Author**

Reviewer #1: All comments have been addressed

Reviewer #2: All comments have been addressed

2. Is the manuscript technically sound, and do the data support the conclusions?

Reviewer #1: Yes

Reviewer #2: Yes

3. Has the statistical analysis been performed appropriately and rigorously?

Reviewer #1: Yes

Reviewer #2: Yes

4. Have the authors made all data underlying the findings in their manuscript fully available?

Reviewer #1: Yes

Reviewer #2: Yes

5. Is the manuscript presented in an intelligible fashion and written in standard English?

Reviewer #1: Yes

Reviewer #2: Yes

Reviewer #1: The authors responded to my comments promptly, and the manuscript has been revised accordingly. I recommend the revised manuscript for publication.

Reviewer #2: Author have responded correctly to my comments in detail.

This study has reported novel and clinically applicable findings based on cutting-edge and accurate analytical methods, and can provide new insights into the development of biomarkers and novel treatments for IPF.

**Do you want your identity to be public for this peer review?** For information about this choice, including consent withdrawal, please see our Privacy Policy

Reviewer #1: **Yes: ** Toyoshi Yanagihara

Reviewer #2: No

---

## [Editor Report · Acceptance letter]

PONE-D-25-20083R1

PLOS ONE

Dear Dr. Yu,

I'm pleased to inform you that your manuscript has been deemed suitable for publication in PLOS ONE. Congratulations! Your manuscript is now being handed over to our production team.

Kind regards,

on behalf of

Dr. Yoshiaki Zaizen

Academic Editor

PLOS ONE